# Fus3, as a Critical Kinase in MAPK Cascade, Regulates Aflatoxin Biosynthesis by Controlling the Substrate Supply in *Aspergillus flavus*, Rather than the Cluster Genes Modulation

Longxue Ma,[a] Xu Li,[a] [ORCID] Fuguo Xing,[a] Junning Ma,[a] Xiaoyun Ma,[a] Yiran Jiang[a]

[a]Institute of Food Science and Technology, Chinese Academy of Agricultural Sciences/Key Laboratory of Agro-products Quality and Safety Control in Storage and Transport Process, Ministry of Agriculture, Beijing, People's Republic of China

Longxue Ma and Xu Li contributed equally to this article. Author order was determined based on investigation, methodology, data curation, project administration, and writing - original draft.

**ABSTRACT** The Fus3-MAP kinase module is a conserved phosphorylation signal system in eukaryotes that responds to environmental stress and transduction of external signals from the outer membrane to the nucleus. *Aspergillus flavus* can produce aflatoxins (AF), which seriously threaten human and animal health. In this study, we determined the functions of Fus3, confirmed Ste50-Ste11-Ste7-Fus3 protein interactions and phosphorylation, and explored the possible phosphorylation motifs and potential targets of Fus3. The regulatory mechanism of Fus3 on the biosynthesis of AF was partly revealed in this study. AF production was downregulated in Δ*fus3*, but the transcriptional expression of most AF cluster genes was upregulated. It is notable that the levels of acetyl-CoA and malonyl-CoA, the substrates of AF, were significantly decreased in *fus3* defective strains. Genes involved in acetyl-CoA and malonyl-CoA biosynthesis were significantly downregulated at transcriptional or phosphorylation levels. Specifically, AccA might be a direct target of Fus3, which led to acetyl-CoA carboxylase activities were decreased in null-deletion and site mutagenesis strains. The results concluded that Fus3 could regulate the expression of acetyl-CoA and malonyl-CoA biosynthetic genes directly or indirectly, and then affect the AF production that relies on the regulation of AF substrate rather than the modulation of AF cluster genes.

**IMPORTANCE** *Aspergillus flavus* is an important saprophytic fungus that produces aflatoxins (AF), which threaten food and feed safety. MAP (mitogen-activated protein) kanases are essential for fungal adaptation to diverse environments. Fus3, as the terminal kinase of a MAPK cascade, interacts with other MAPK modules and phosphorylates downstream targets. We provide evidence that Fus3 could affect AF biosynthesis by regulating the production of acetyl-CoA and malonyl-CoA, but this does not depend on the regulation of AF biosynthetic genes. Our results partly reveal the regulatory mechanism of Fus3 on AF biosynthesis and provide a novel AF modulation pattern, which may contribute to the discovery of new strategies in controlling *A. flavus* and AF contamination.

**KEYWORDS** aflatoxin biosynthetic regulation, *Aspergillus flavus*, Fus3-MAPK cascade, phosphorylation, acetyl-CoA carboxylase

Address correspondence to Fuguo Xing, xingfuguo@caas.cn.

The authors declare no conflict of interest.

*Aspergillus flavus* is a ubiquitous saprophytic and pathogenic fungus that infects several important agricultural crops, such as maize (ear rot), peanut (yellow mold), and cottonseed in pre- or postharvest (1), and produces diverse toxic chemicals posing a huge health risk worldwide, especially aflatoxins (AF) (2–4). AF are a class of hypertoxic and carcinogenic benzofuran compounds that are mostly produced by *Aspergillus* section *Flavi* (5). Among them, aflatoxin B1 (AFB$_1$) is the most critical

because of its frequent presence in food and feeds and severe acute and chronic toxicity to humans and livestock (6, 7).

The biosynthesis process and regulatory mechanisms of AF have been well studied. The acetyl-CoA and malonyl-CoA, produced by primary metabolism, are regarded as AF precursors. A 75-kb AF cluster, located on the 3# chromosome telomere, is responsible for AF biosynthesis (8). The pathway-specific regulators, AflR and AflS, oxidative stress transcription factors (TFs), such as AtfA, AtfB, SrrA, and AP-1, and several global regulators, jointly participate in the regulations of AF cluster genes (1, 9–12). In addition, different environmental factors stimulate these regulators and AF genes and subsequently control AF biosynthesis or other metabolisms that rely on transcriptional and post-transcriptional regulation (13). During these processes, phosphorylated modification with the mitogen-activated protein kinase (MAPK) pathway is one of the most critical regulatory systems, and the MAPK modules are involved in diverse metabolisms, including colonization, mating, differentiation, development, conidiation, as well as AF biosynthesis (14).

MAPK cascades consist of three conserved kinases: MAP kinase kinase kinase (MAPKKK), MAP kinase kinase (MAPKK), and MAP kinase (MAPK) (13–16). The MAPKKK is activated by the G-protein coupled receptor (GPCR), which receives and transmits extracellular signals, that activate one another sequentially via phosphorylation, and then the activated MAPK phosphorylates downstream targets (17). Depending on the terminal kinases, different MAPK signaling pathways, such as SakA-MAPK, Slt2-MAPK, and Fus3-MAPK, are identified in fungi, and the Fus3-MAPK cascade receives more attention because of its crucial functions and its wide regulations (13, 14, 18). In *Saccharomyces cerevisiae*, the central complex of Ste11 (MAPKKK), Ste7 (MAPKK), and Fus3 (MAPK) are assembled on the scaffold protein Ste5, and their close proximity enhances the transmission of phosphorylation and the flow of information (19). In *A. nidulans*, the Fus3-MAPK modules are composed of Fus3, the upstream kinases Ste7, Ste11, and the adaptor Ste50, but only Fus3 has the potential to enter the nucleus and phosphorylate the sexual development regulator Ste12 (SteA) and the global regulator VeA (14). Similarly, in *A. flavus*, the tetrameric complex, consisting of three kinases, Ste11 (SteC), Ste7 (MkkB), and Fus3 (MpkB), and the adaptor Ste50 (SteD), is assembled in the cytoplasm and the phosphorylated Fus3 can translocate into the nucleus (20). Therefore, Fus3, as the terminal kinase of the Fus3-MAPK cascade, phosphorylates downstream targets and then influences diverse phenotypes in *Aspergillus*.

The functions of Fus3 on sexual and asexual development have been described before (14, 21). Deletion of *fus3* leads to the inhibition of hyphal extension and differentiation, and abolishment of asexual sporulation and sclerotia formation in diverse *Aspergillus* species (22). The main regulatory mechanisms are that Fus3 directly interacts with the velvet complex and activates VeA and Ste12 by phosphorylated modification, and these two global regulators coordinate fungi development (14, 23). Fus3 is also essential for secondary metabolism in *Aspergillus*. The sterigmatocystin in *A. nidulans* and the gliotoxin in *Aspergillus fumigatus* are affected by Fus3 (14, 24, 25). Notably, Yang et al. and Frawley et al. indicated that Fus3 might be a positive regulator of AF in *A. flavus* (13, 20). However, the regulatory mechanism of Fus3 in AF biosynthesis is still unclear.

Here, we performed a comprehensive and in-depth study on Fus3 in *A. flavus*. Physical interaction and phosphorylation transmission were detected with yeast two-hybrid (Y2H), tandem affinity purification (TAP), and Western blotting. Potential targets of Fus3 were identified in phosphoproteome analysis and the putative motif of Fus3 targets was identified as [RxxSP]. More importantly, we partly revealed the regulatory mechanism of Fus3 on AF biosynthesis. Our results indicate that Fus3 could affect AF biosynthesis by controlling the AF substrate supply, but not by regulating AF gene expression. These data highlight a novel regulatory mechanism involving the MAPK cascade and secondary metabolism in *Aspergillus*.

## RESULTS

**Fus3 positively regulates fungal development and AFB$_1$ production in *A. flavus*.** In this study, *fus3* null-deleted and complementary strains were generated (Fig. 1 and

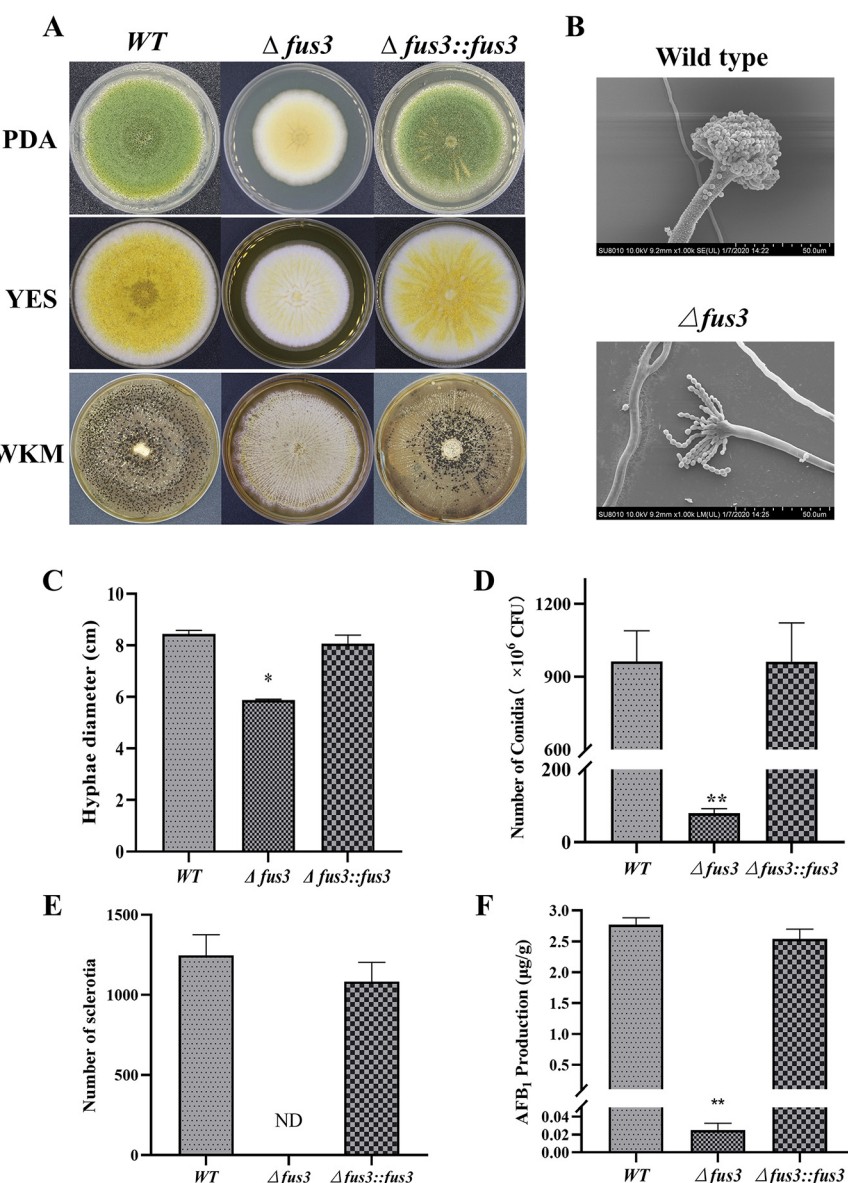

**FIG 1** Phenotypes of Δ*fus3* and complementary strains. (A) WT, Δ*fus3*, and complementary strains (Δ*fus3*::*fus3*) were cultured on PDA, YES, and WKM plates; (B) the maldevelopment and sterility of conidia head and conidiophore of *fus3* deletion under the scanning electron microscope; (C) the growth rate; (D) the conidia number, (E) the sclerotia number, and (F) the AFB$_1$ production of WT, Δ*fus3*, and complementary strains. ND, not detected; * and ** show a significant difference at $P < 0.05$ and $P < 0.01$, respectively.

Fig. S1). Deletion of *fus3* led to obvious impairment of fungal development, such as a decrease in mycelia growth (Fig. 1A and C) and the maldevelopment and sterility of conidia head and conidiophore (Fig. 1B), and a significant reduction in conidia production (Fig. 1D). Additionally, the sclerotia yield of Δ*fus3* was severely decreased compared with the wildtype (WT) in Wickerham (WKM) media (Fig. 1A and E). AFB$_1$ production was also significantly decreased in the *fus3* mutant (Fig. 1F). All results concluded that Fus3 is a critical kinase that could positively regulate mycelia growth, conidia development, sclerotia formation, and AFB$_1$ biosynthesis. The mutants of other MAPK genes (*ste7*, *ste11*, and *ste50*) were also generated in this study, and the phenotypes of these mutants were similar to that of Δ*fus3* (Fig. S1), implying that these MAPK modules could affect *A. flavus* development and metabolism depending on the Fus3.

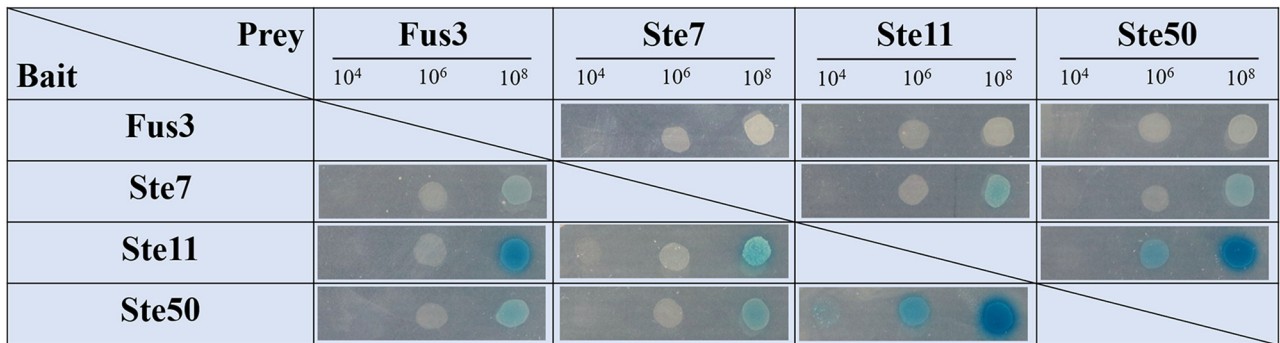

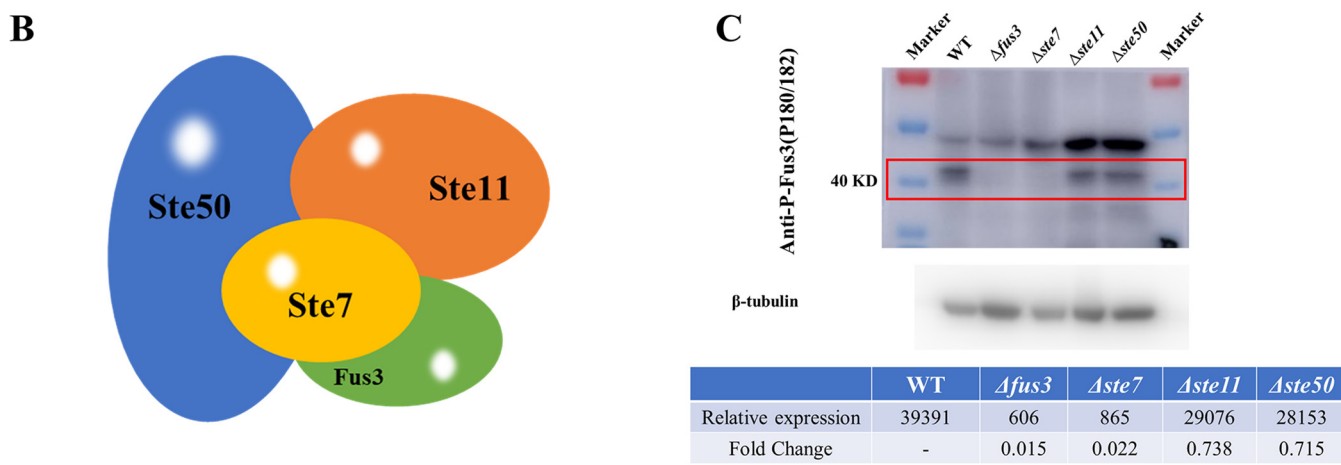

**FIG 2** The interaction and phosphorylation of Fus3-MAPK pathway. (A) The interactions between MAPK pathway proteins were detected by Y2H; (B) the interaction model of MAPK modules; (C) the phosphorylation of Fus3 was detected by Western blotting with the p-P38 antibody, and the p-Fus3 protein is about 42 KD (the relative expression value was calculated with ImageQuant TL); (D) the interactions between the site mutagenesis Fus3 and the other MAPK modules.

**Interaction and phosphorylation of Fus3 in MAPK cascade proteins.** To investigate and verify the phosphorylation transmission of the MAPK pathway in *A. flavus*, Y2H, TAP, and Western blotting analyses were performed. As Fig. 2A shows, Fus3 strongly interacts with Ste11, moderately interacts with Ste50, and slightly interacts with Ste7. Ste7 showed moderate and slight interactions with Ste11 and Ste50, whereas Ste11 and Ste50 had a strong interaction (Fig. 2A). We also confirmed interactions by Fus3-TAP analysis (Fig. S2 and Table S3). A total of 862 proteins were identified by liquid chromatography-mass spectrometry/mass spectrometry (LC-MS/MS) and regarded as the potential direct interaction targets of Fus3 (Table S3). In addition to these MAPK modules, 20 kinase proteins were observed (Table S3), implying a complicated cross talk of phosphorylation regulation in *Aspergillus*. Based on these interaction results and previous research, the interaction model of MAPK tetrameric complex in *A. flavus* was predicted. Ste50 as the holder could stabilize other MAPK members, and Ste11-Ste7-Fus3 could act as a trimer, transferring the phosphorylation signal along Ste11, Ste7, and Fus3 (Fig. 2B).

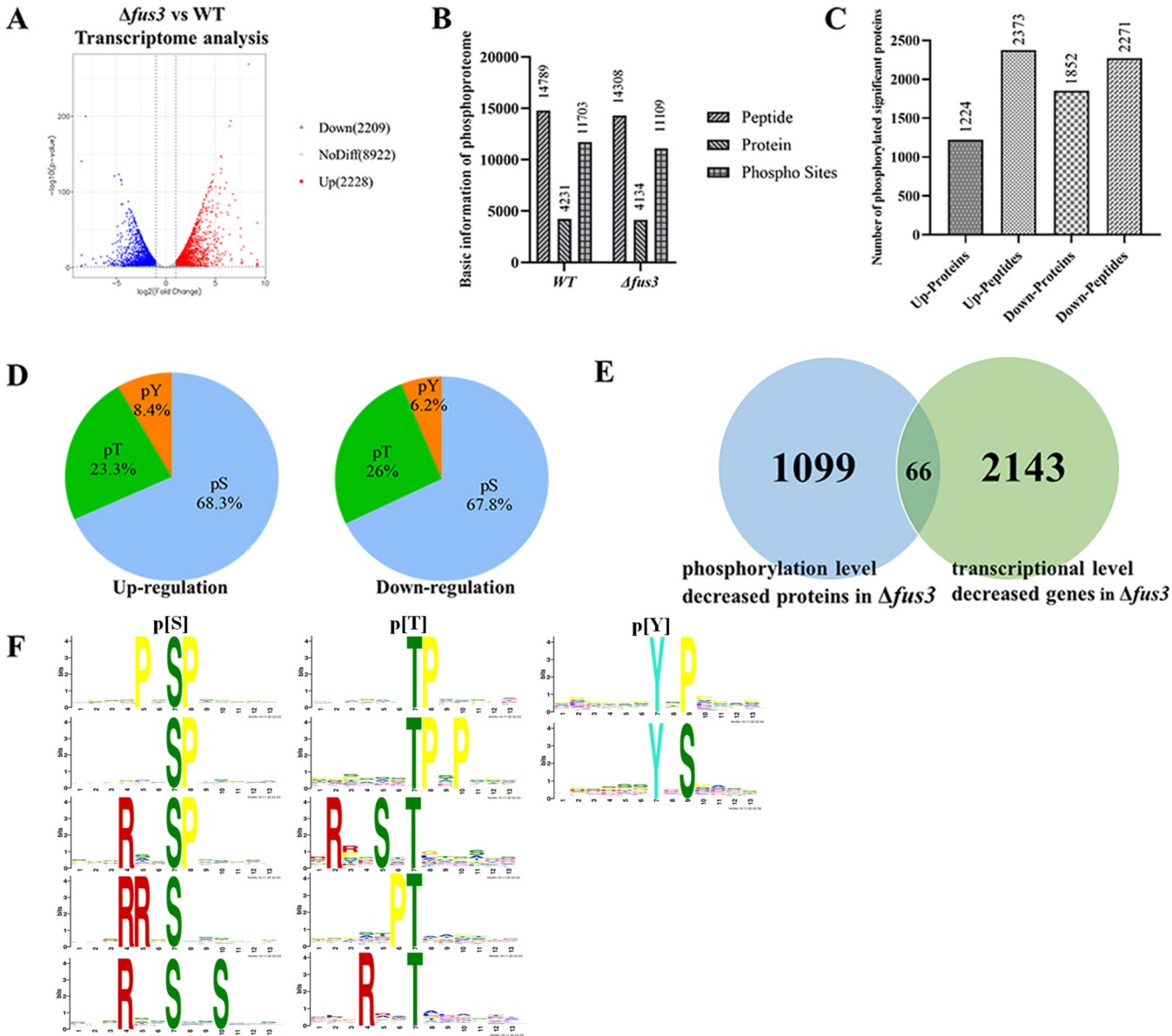

**FIG 3** Transcriptome and phosphoproteome analyses of WT and Δfus3. (A) The volcano graph shows the number of differentially expressed genes in Δfus3 versus WT; (B) the number of phosphorylated peptides, phosphorylated sites and phosphorylated proteins in Δfus3 and WT; (C) the number of proteins and peptides with upregulated or downregulated phosphorylations in Δfus3 versus WT; (D) pie charts showed the distributions of different phosphorylation site in up- and downregulated phosphorylation in Δfus3; (E) Venn diagram illustrates the downregulated genes in transcriptional level and phosphorylation levels in Δfus3; (F) the top five prevalent phosphorylation motifs from differentially phosphorylated peptides.

The level of Fus3 phosphorylation was examined in different mutants, and 42 kD-Fus3 bands were detected by p38-phosphorylated antibody (Fig. 2C). No visible bands emerged in Δfus3 and Δste7, and weak bands were noticed in Δste11 and Δste50, with a 0.738- and 0.715-fold decrease compared with WT, respectively (Fig. 2C). All these results suggest that the phosphorylation level of Fus3 could be affected by three MAPK modules, and Ste7 might directly phosphorylate Fus3.

**Comprehensive analyses of transcriptome and phosphoproteome in Δfus3.** To investigate the functions and the target of Fus3, comprehensive transcriptome and phosphoproteome analyses were performed between *A. flavus* WT and Δfus3. Via RNA-seq, 59.91 million and 72.94 million clean reads were obtained after filtering in *A. flavus* WT and Δfus3, respectively. Overall, 4,437 differentially expressed genes (DEGs) were recognized in the *fus3* deletion strain (false-discovery rates [FDR] ≤ 0.05, $\log_2$FC ≥1 or ≤ −1), with 2,209 downregulated and 2,228 upregulated (Fig. 3A).

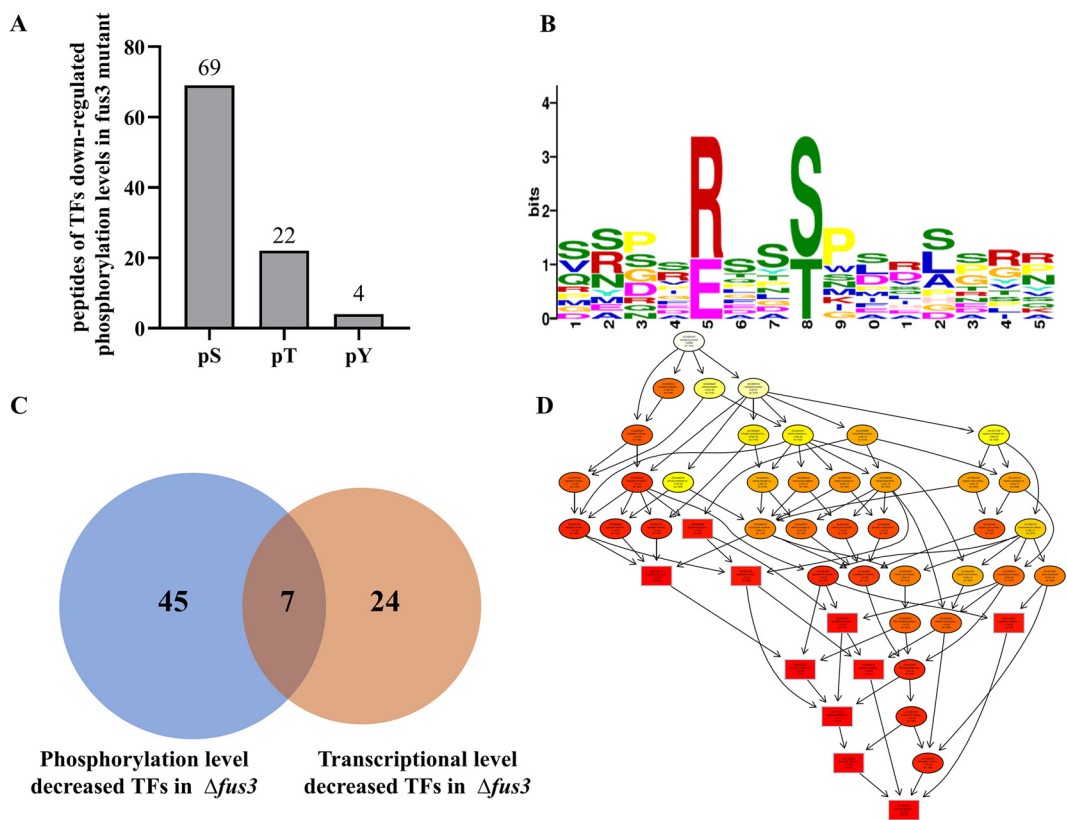

**FIG 4** Analysis of 45 potential target transcription factors of Fus3. (A) The bar graph shows the number of diverse phosphorylation site in the downregulated phosphorylation peptides; (B) the predicted phosphorylation motif of the Fus3 potential targets; (C) Venn diagram illustrates the downregulated TFs in transcriptional level and phosphorylation levels in Δ*fus3*; (D) the top 10 biological process in GO term of the possible target of Fus3.

Using an mass spectrum (MS)-based high-throughput proteomic approach, 11,703 phosphorylation sites in 4,231 proteins and 11,109 phosphorylation sites in 4,134 proteins were identified in WT and Δ*fus3*, respectively (Fig. 3B). In Δ*fus3*, 2,373 phosphorylated peptides in 1,224 proteins were significantly upregulated and 2,271 phosphorylated peptides in 1,852 proteins were significantly downregulated (Fig. 3C). Phosphorylated sites (phosphoserine [pS], phosphothreonine [pT], and phosphotyrosine [pY]) included 68.3% pS, 23.3% pT, and 8.4% pY in upregulated phosphorylation peptides, and 67.8% pS, 26.0% pT, and 6.2% pY in significantly downregulated peptides (Fig. 3D).

To reduce the interferences from the decrease in mRNA level, a total of 1,033 downregulated phosphorylation proteins in Δ*fus3* were regarded as potential Fus3 targets, excluding 66 proteins with significant downregulation at the transcriptional level (Fig. 3E). Enriched motifs of these Fus3 potential targets were analyzed with motif X (24). The most prevalent motifs are listed in Fig. 3F. Among the top five popular motifs of pS, the novel motif [PxSP] was the most prevalent with the highest match score, and [xTPx] was the most prevalent motif in pT (Fig. 3F). Only two motifs, [xxYxP] and [xxYxS], were predicted in pY analysis (Fig. 3F).

**Fus3 affects the phosphorylation levels of diverse TFs.** Because *fus3* deletion leads to a lot of transcriptional variations, it is reasonable to predict that Fus3 could regulate several TFs at the phosphorylation level, and then affect transcription of genes downstream. Based on phosphoproteome data, 95 peptides (with 69 pS, 22 pT, and 4 pY) in 52 TFs showed significantly downregulated phosphorylation in Δ*fus3* (Fig. 4A). Especially with AtfA, seven different downregulated peptides of AtfA were discovered in Δ*fus3* (Table S2). The predicted phosphorylation motif of these peptides was [R/

ExxS/Txx] in pS/T (Fig. 4B). Two significantly downregulated phosphorylated peptides in SteA, [RSATMME] and [RHASMPT], were matched to the motif.

A total of 45 TFs were downregulated at the phosphorylation level, but not downregulated at the transcriptional level (Fig. 4C). Among them, AFLA_025030 (Hsf1), SrrA, AFLA_033160 (Sfp1), AFLA_086590 (Smp1), and AP-1 were involved in stress response; NsdD, SteA, AFLA_030600 (Fkh1), AFLA_078390 (Sin3), Con7, AFLA_088390 (Btf3), and AFLA_133380 were involved in fungal development and differentiation; AFLA_017900, AFLA_029340, and AFLA_054810 mainly responded to carbon sources, and two regulators, AreA and AreB, participated in nitrogen metabolism (Fig. 4D, Table S2). Therefore, these 45 TFs might be direct targets of Fus3 and regulate the transcription of downstream genes. VeA, reported as a direct target of Fus3, was also identified as a significantly downregulated phosphorylated peptide (GPAPAAV**S**TPAPPAP) in our study.

**Effects of Fus3 on conidial genes by transcriptome and phosphoproteome.** To explain lower conidia production and abnormal conidial development in Δ*fus3*, the expression of several conidia developmental genes were analyzed (Fig. 5D). In Δ*fus3*, six conidial developmental genes, including conidiation-specific family protein (AFLA_044790), conidiation proteins *con6* and *con10*, and conidial hydrophobin, *rodA* has a significantly downregulated transcriptional level. The phosphorylation levels of these conidial proteins were not significantly changed (Fig. 5D). The transcriptional expression of *vosA*, *flbC*, and *wetA* were downregulated in Δ*fus3* (Fig. 5D). Con7, a specific conidia development TF, had no significant variation at the transcriptional level but had significantly decreased phosphorylation (Fig. 5D). The qPCR results were similar to the transcriptome data, except for the upregulation of *brlA* (Fig. 5G). Taken together, Fus3 might be a critical kinase in conidia development, which could affect the expression of conidia developmental genes and their regulators at transcriptional and phosphorylation levels.

**T182 and Y184 are the critical phosphorylation sites of Fus3.** Based on phosphoproteomics, the T182 and Y184 residues were the predicted phosphorylated sites of Fus3 in *A. flavus* NRRL3357 (177-NSGFM**TEY**VATR1-188), accordant with previous studies (13, 24). Site-mutagenesis was used to generate strains with T182 and Y184 replaced by alanine (A) and asparagine (D), respectively (Fig. S2D). Similar to Δ*fus3*, the mutagenesis strains were defective in mycelia growth, conidia production, and sclerotia formation (Fig. 6A and C). The AF production of the four site-mutated strains was significantly decreased compared with WT, but still higher than Δ*fus3* (Fig. 4B). All these results suggest that these phosphorylation sites are essential for the regulation of Fus3.

The interactions of Fus3 and the other MAPK modules were also affected by these phosphorylated sites. Compared with native Fus3, Fus3[T182A], Fus3[Y184A], and Fus3[Y184D] showed slightly enhanced interactions with Ste7, Ste11, and Ste50, respectively (Fig. 2D). But the interactions between Fus3[T182D] and the other MAPK modules were significantly decreased (Fig. 2D), suggesting that Fus3[T182] might be the critical interaction site, and the replacement threonine with asparagine might obstruct its interactions.

**Fus3 regulates AF production independently of AF biosynthetic genes.** To reveal the regulatory mechanisms of Fus3 on AFB$_1$ biosynthesis, the expression of the AF biosynthetic genes and relative regulators were examined by transcriptomic and phosphoproteomic analyses. Although AFB$_1$ production in Δ*fus3* was significantly downregulated (Fig. 1F), most of (25/33) the AF cluster genes showed a significant increase at the transcriptional level (Fig. 5A). The cluster-specific regulators, AflR and AflS, increased by 2.00- and 2.89-log$_2$FC in Δ*fus3* than WT, respectively (Fig. 5A). Similarly, most AF biosynthetic genes (9/13) and *aflS* were significantly upregulated in Δ*fus3*, but *aflR* showed no significant difference in quantitative real-time PCR (RT-qPCR, qRT-PCR, q-PCR) analysis (Fig. 5E). Based on phosphoproteome data, the phosphorylation levels of AF cluster genes were not significantly varied in Δ*fus3* (Fig. 5A).

The velvet complex genes, *veA* and *velB*, showed no obvious differences in transcriptional level, but *laeA* was significantly decreased in Δ*fus3* (Fig. 5E). Several TFs, such as *msnA*, *mtfA*, *creA*, and *areA*, showed significantly increased transcriptional expression in Δ*fus3*, but *atfA* and *atfB* were significantly decreased, with 1.2- and 1.58-log$_2$FC reductions,

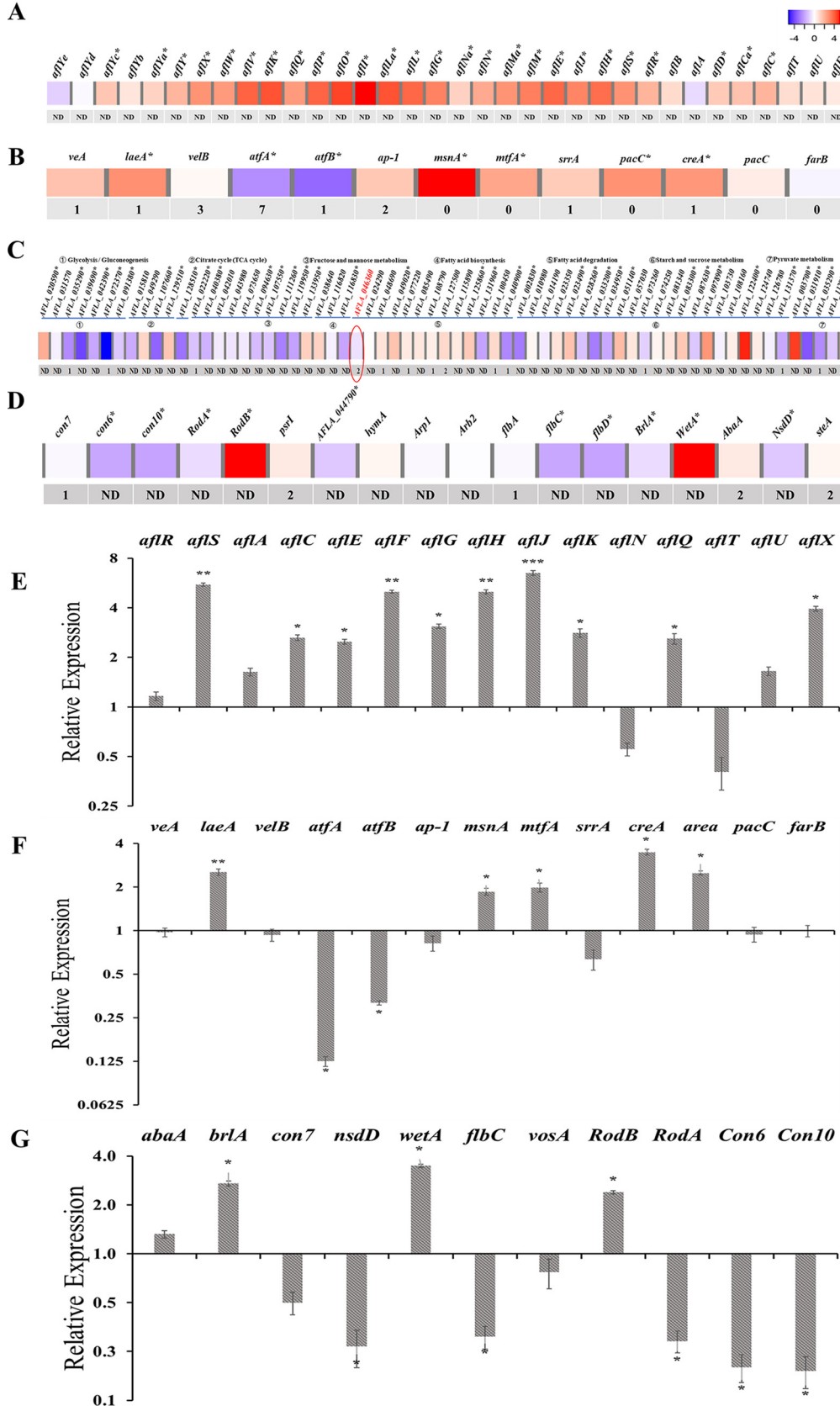

**FIG 5** Transcriptome, phosphoproteome, and RT-qPCR analyses of Δfus3 versus WT. (A) The expressions of AFs biosynthesis cluster genes; (B) the expressions of diverse global regulators; (C) the expressions of genes involved in

respectively (Fig. 5B). In RT-qPCR verification, the transcriptional expressions of *laeA*, *msnA*, *mtfA*, *creA*, and *areA* were significantly upregulated, but *atfA* and *atfB* levels were significantly downregulated in Δ*fus3* (Fig. 5F). More importantly, downregulated phosphorylation peptides of these global regulators were identified in the Δ*fus3* mutant, including VeA, AtfA, AtfB, AP-1, SrrA, and AreA (Fig. 5F), suggesting that the phosphorylation level of these global regulators might be strongly defective in Δ*fus3*.

The above results concluded that Fus3 could affect the expression of AF biosynthetic genes and several regulators. However, the increase in expression of AF cluster genes was not accordant with the phenotype change (a significant decrease of AF production) in Δ*fus3*. Therefore, Fus3 might regulate AF biosynthesis independently of the AF cluster genes.

**Fus3 could affect AF biosynthesis by modulating AF biosynthetic substrates.** Paradoxically, the AF cluster genes were upregulated at the transcriptional level, but AF production was decreased in Δ*fus3*. Thereafter, a novel regulatory mechanism of Fus3 on AF biosynthesis might be involved in this regulation. Acetyl-CoA and malonyl-CoA, the precursor of AF, are also directly relevant to AF production (25, 26). The acetyl-CoA and malonyl-CoA levels were examined by CoA-ester enzyme linked immunosorbent assay (ELISA) kits. The acetyl-CoA levels in Δ*fus3* and the site mutagenesis strains were significantly decreased compared with WT (Fig. 6D). Furthermore, the malonyl-CoA levels of the *fus3* null-deleted strain and the site-mutagenesis strains were also significantly lower than WT (Fig. 6E). Interestingly, the malonyl-CoA levels of the site-mutagenesis strains were higher than Δ*fus3* (Fig. 4F). All results suggested that defective Fus3 could impair acetyl-CoA and malonyl-CoA biosynthesis.

Acetyl-CoA is mostly generated from the glycolysis (afv00010) and fatty acid $\beta$-oxidation (afv00071) pathways. Several genes related to acetyl-CoA were affected by Fus3 in this study. For example, in glycolysis, AFLA_072370, encoding a hexokinase, was significantly downregulated at the mRNA level ($-4.31$-$\log_2$FC) and downregulated at the phosphorylated level. Pyruvate dehydrogenase (AFLA_035290), another key enzyme in the glycolysis pathway, was also suppressed in Δ*fus3* at both the transcriptional level ($-2.15$-$\log_2$FC) and phosphorylation level; the phosphoglycerate mutase (AFLA_039690) and glyceraldehyde 3-phosphate dehydrogenase (AFLA_042390) in Δ*fus3* showed $-3.42$ and $-1.63$ $\log_2$FC reductions at the transcriptional level, respectively (Fig. 5C). Only two genes, AFLA_131960 and AFLA_040900 with $-1.41$ and $-2.23$ $\log_2$FC, respectively, in the fatty acid degradation pathway (afv0071), were significantly reduced in Δ*fus3*, but seven proteins in fatty acid degradation were significantly decreased at the phosphorylation level. Additionally, two genes in the citrate cycle, two genes in fructose and mannose metabolism, one gene in fatty acid biosynthesis, two genes in fatty acid degradation, seven genes in starch and sucrose metabolism, and three genes in pyruvate metabolism were significantly downregulated at the transcriptional level (Table S1). A total of 111 proteins involved in carbon metabolism were identified in Fus3-TAP analysis (Table S3). Taken together, Fus3 could affect the expression of several carbon metabolism genes at the transcriptional and post-translational levels and then affect the acetyl-CoA level.

The acetyl-CoA carboxylase (ACCase) gene (*accA*, AFLA_046360) is responsible for the transformation from acetyl-CoA to malonyl-CoA. Based on transcriptome and phosphoproteome analyses, the expression of *accA* had no significant variation at the transcriptional level, but the phosphorylation level of AccA was significantly downregulated in Δ*fus3* (Fig. 5C). In addition, AccA was identified by Fus3-TAP analysis

**FIG 5** Legend (Continued)

carbon metabolism; (D) the expressions of genes involved in fungal development. The red cubes mean upregulated expressions in transcriptome analysis, and the bule cubes mean downregulated. * means a significant difference at $P < 0.05$. The numbers under the cubes stand for the amount of downregulated phosphorylated peptides in the protein. ND, not detected. The expressions of (E) AFs biosynthesis genes, (F) the global regulators, and (G) the conidia developmental genes were examined by RT-qPCR analysis in Δ*fus3* versus WT. Three independent biologic replicates were performed, and data was presented as means ± SD. *t* tests were applied for significance analyses with *, $P < 0.05$; **, $P < 0.01$; ***, $P < 0.001$.

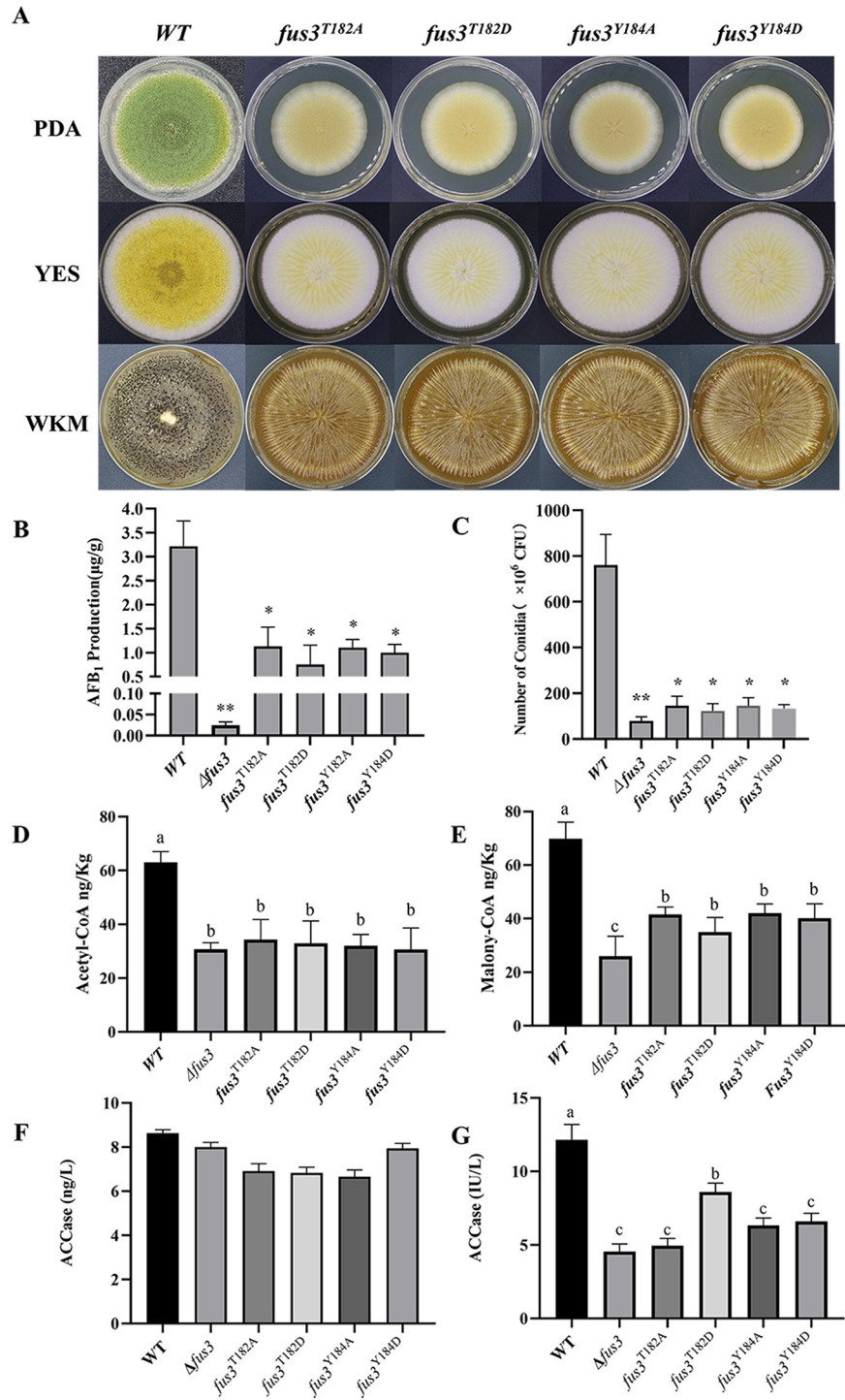

**FIG 6** The variations of development, AFs production, AFs substrates levels, and ACCase activities in *fus3* defective strains. (A) WT and mutagenesis strains (*fus3*$^{T182A}$, *fus3*$^{T182D}$, *fus3*$^{Y184A}$, *fus3*$^{Y184D}$) were cultured on PDA, YES, and WKM agar plates; the AFB$_1$ production (B) and the conidia production (C) of WT and *fus3* defective strains; (D) the acetyl-CoA levels in WT, Δ*fus3*, and four site-mutagenesis strains; (E) the malonyl-CoA levels in WT, Δ*fus3*, and four site-mutagenesis strains. (F) the ACCase quantities in WT, Δ*fus3*, and four site-mutagenesis strains; (G) the ACCase activities in WT, Δ*fus3*, and four site-mutagenesis strains. * and ** show a significant difference at $P < 0.05$ and $P < 0.01$, respectively, and different letters marked the significant difference at $P < 0.05$.

(Table S3), suggesting that AccA might be a direct interaction protein of Fus3. Interestingly, in the analysis of the AccA amino acid sequence, the [1836-RKG**S**PNP-1844] peptide in AccA matched [RxxSP], which is the predicted target motif of Fus3. All this data implies that AccA might be the direct target of Fus3.

We also detected the ACCase quantities and ACCase activities of WT, Δ*fus3*, and the site-mutagenesis strains (Fig. 6F and G). The ACCase quantities of Δ*fus3* and the site-mutated strains were decreased compared with WT, but without significant difference (Fig. 6F). The ACCase activity was significantly impaired in Δ*fus3*. In the four site-mutated strains, ACCase activities were lower than WT, but higher than Δ*fus3* (Fig. 6G). Among the site-mutagenesis strains, *fus3*$^{T182A}$ showed the lowest ACCase activity, but *fus3*$^{T182D}$ partly recovered the ACCase activity (Fig. 6G). Therefore, T182, a critical phosphorylation site in Fus3, is involved in the ACCase activity changes, implying the Fus3 phosphorylation level is highly correlated with ACCase activity. Taken together, Fus3 could regulate the phosphorylation level of AccA directly, then modulate the ACCase activity, and subsequently affect malonyl-CoA biosynthesis and AF production.

## DISCUSSION

The MAPK pathway, consisting of a cascade of three protein kinases, represents a highly conserved eukaryotic signal transduction system from yeast to humans (27, 28). In fungi, the outer membrane signals are detected by GPCRs and activate the MAPK modules (29–31). The phosphorylated and activated MAPK modules regulate the downstream TFs directly or indirectly (13, 14). Therefore, interaction and phosphorylation are critical for MAPK signal transduction. In *A. nidulans*, five proteins, comprising three kinases SteC (Ste7), MkkB (Ste11), and MpkB (Fus3), the adaptor protein SteD, and the scaffold HamE, co-accomplish the phosphorylate signal transfer, whereas, in *A. flavus*, Fus3-MAPK modules consist of the three kinases and Ste50 (20). It is speculated that Ste50 might function as both the scaffold protein and the adaptor protein. In fact, we noticed obvious interactions between Ste50 and the other three kinases, and the Fus3 phosphorylation level in Δ*ste50* was significantly decreased (Fig. 2A and C), implying that Ste50 should play a crucial role in the interaction and phosphorylation of the MAPK system. A previous study reported that the Ste7-Fus3 localizes in the cytoplasm as a dimer (32), but in our study, the interaction of Ste7 and Fus3 was weaker than the interaction of Ste11 and Fus3, and the Fus3-TAP result confirmed the interaction of Fus3 and Ste11, implying that Ste11-Ste7-Fus3 could be a triad protein complex (Fig. 2A). Furthermore, no Fus3-p was observed in Δ*ste7*, but Fus3-p was observed in Δ*ste11* (Fig. 2C), suggesting that only Ste7 could directly phosphorylate Fus3, but Ste7 could be phosphorylated by more than Ste11. Several upregulated kinases were identified by Δ*fus3*-phosphoproteome and Fus3-TAP, and the cross talk between diverse MAPK cascades and kinases are similar in Eukarya (32–34).

The MAPK kinase is phosphorylated at the conserved tripeptide T-X-Y (31, 35). Consistently, we identified the [182-T-E-Y-184] motif as the phosphorylation site of Fus3. The phenotypes of all site-mutagenesis strains were similar to the *fus3* null-deletion mutant, suggesting these two sites might be essential for Fus3 functions. Interestingly, Fus3$^{T182A}$, Fus3$^{Y184A}$, and Fus3$^{Y184D}$ normally interact with other proteins, whereas hardly any interaction was noticed between Fus3$^{T182D}$ and the other MAPK modules (Fig. 2D). A possible explanation is that the 182T site might be a critical interaction site of Fus3, and the replacement of aspartic acid might interfere with its structure and interaction. Phosphorylation is the predominant post-translational modification in cellular signaling. Fus3, as the terminal kinase of the Fus3-MAPK cascade, could modify several proteins by phosphorylation (36). Among the differential phosphorylated proteins, half showed an increased phosphorylation level (1,260 upregulated and 1,165 downregulated phosphorylated proteins in Δ*fus3*), which also demonstrated cross talk in different phosphorylation pathways. Once a pathway is blocked, the other kinase systems will be upregulated to rescue it (37, 38).

To find the potential targets of Fus3, we focused on 1,033 downregulated phosphorylation

level proteins (Fig. 3E). Of these, intracellular signal transduction (GO: 003556) and DNA-binding transcription factor activity (GO:0003700) were the most prevalently enriched gene ontology (GO) terms (Data not shown). The enriched motifs of the Fus3 potential targets were identified. For pS, the most prevalent motif [PxSP] might be a novel motif in kinases. [S/TP] is potentially recognized by the kinases of MAPK and ERK pathways (39). [RxxSP] is a typical cyclin-dependent kinase (CDK) target motif (38). [RRxS], [RxxS/T], and [RxxSxxS] are recognized as cAMP-dependent kinase (PKA) targets (40, 41). Furthermore, [R/ExxS/T] was identified as a potential Fus3 motif in diverse TFs of *A. flavus*. In consideration of the high probability of proline after the phosphorylation site (S/T) (Fig. 4B), we suggested that [RxxSP] might be the most possible target motif of Fus3 in *A. flavus*. In addition, we also identified the [RxxSP] motif in AccA, which further confirmed that AccA might be a potential target of Fus3.

In this study, a large number of genes were changed at the transcriptional level in Δ*fus3* (Fig. 3A), implying that the various TFs might be affected by Fus3. In fact, 45 TFs were significantly downregulated at the phosphorylation level and identified as the potential targets of Fus3 in *A. flavus* (Table S2). Several studies have reported that the Fus3-MAPK cascade strictly regulates both asexual and sexual development in *Aspergillus*. Deletion of *fus3* almost abolished sclerotia formation, in *A. nidulans*, *A. fumigatus*, *A. niger*, and *A. flavus*, and severely damaged the conidia production in *A. nidulans* and *A. flavus* (14, 20, 37, 42). Fus3 could regulate fungal development by phosphorylated modification of VeA and SteA in *Aspergillus* strains (14), and we also confirmed that VeA and SteA are the potential targets of Fus3 in this study. Apart from VeA and SteA, the phosphorylation levels of NsdD and Con7, which are responsible for sexual and asexual development, respectively (42, 43, 44), were also significantly decreased in Δ*fus3* (Fig. 5). These phosphorylation level variations of these regulators would lead to the transcriptional changes of downstream genes. For example, the transcriptional expressions of the conidiation genes, *con6*, *con10*, and *rodA*, were significantly downregulated at transcriptional levels in Δ*fus3* (Fig. 5G). The phenotypes of the site-mutagenesis strains also illustrated that the Fus3 phosphorylation is indispensable for sexual and asexual development (Fig. 6A). Therefore, Fus3 could directly phosphorylate the developmental TFs, then regulate the transcriptional level of downstream developmental genes, and subsequently affect the sexual and asexual development of fungi. Furthermore, apart from developmental TFs, diverse TFs were also involved in the phosphorylated regulation of Fus3, such as oxidative stress TFs, SrrA and AP-1 (45, 46), heat shock TF Hsf1 (47), starvation stress-related TF Sfp1 (48), osmotic stress-related TF Smp1 (49), and nitrogen metabolism TFs, AreA and AreB (Table S2). Overall, Fus3 is a global kinase that could phosphorylate many regulators, participate in various metabolism processes, and affect diverse phenotypes, including cell differentiation, spore formation, stress response, nutrition metabolism, as well as secondary metabolite biosynthesis.

Secondary metabolites of *A. niger*, *A. fumigatus*, *A. nidulans*, and *Fusarium graminearum* are drastically decreased in *fus3*-deleted strains (14, 33, 42, 50). Similar to our results, for the regulation of AF in *A. flavus*, both Frawley et al. and Yang et al. concluded that Fus3 is indispensable for AF biosynthesis (13, 20). Furthermore, the site mutagenesis strains also showed significantly decreased $AFB_1$ levels (Fig. 6B), suggesting that native Fus3 is also necessary for AF biosynthesis. However, no relevant investigations have revealed the regulatory mechanism of Fus3 in AF biosynthesis. In this study, we examined the AF cluster gene expressions in Δ*fus3*. It was unexpected that almost all genes in the AF cluster, including pathway-specific regulators, *aflR* and *aflS*, were significantly upregulated at the transcriptional level in Δ*fus3* (Fig. 5A and E). It is paradoxical that AF cluster genes were upregulated at the transcriptional level in Δ*fus3* but AF production was decreased. Therefore, Fus3 affects AF biosynthesis independently of the regulation of the AF cluster. We considered whether the level of AF biosynthetic substrates, acetyl-CoA and malonyl-CoA, were changed in *fus3* defective strains (1, 51). Crucially, both acetyl-CoA and malonyl-CoA levels in the *fus3* mutant and site-

mutated strains were significantly downregulated (Fig. 4E and F), suggesting that *fus3* defection could directly block the substrate supplement of AF biosynthesis.

Acetyl-CoA originates from carbon metabolism and fatty acid metabolism. Based on the transcriptome data, several genes that are involved in acetyl-CoA formation and accumulation were downregulated (Table S1). In addition, the acetyl-CoA carboxylase gene (AccA, AFLA_046360), responsible for malonyl-CoA synthesis, was significantly downregulated at the phosphorylation level in Δ*fus3* (Fig. 5C), and the ACCase activities in *fus3* defective strains were downregulated (Fig. 6G). Taken together, this suggests that Fus3 could affect the AccA phosphorylation level, and then regulate its activity. Furthermore, AccA was identified by Fus3-TAP analysis (Table S3), and the putative Fus3 phosphorylation motif was discovered in AccA, implying that Fus3 could directly interact with AccA. In yeast, ACCase activity is regulated by phosphorylation and is directly phosphorylated by the protein kinase Snf1 (52). However, Jin et al. indicated that AccI phosphorylation level in *S. cerevisiae* was negatively correlative to its activity, which is contrary to our result in *A. flavus* (53). More investigations about the phosphorylated regulation of ACCase could be performed in future research. Taken together, we conclude that Fus3 could regulate the expression of carbon metabolism genes at the transcriptional and phosphorylation level, especially AccA, then control the levels of AF biosynthetic precursors, acetyl-CoA and malonyl-CoA, and subsequently affect AF production.

In addition, three carbon metabolism regulators, FacB, Ngg1, and RcoA, were downregulated at the phosphorylation level in Δ*fus3* (Table S2), and several carbon metabolism-related genes were observed in Fus3-TAP analysis (Table S3), implying that Fus3 might be deeply involved in carbon metabolism in *A. flavus*. Indeed, the *fus3* mutant was defective in the response to glucose and sugar (54). This data also indicates that Fus3 could affect AF biosynthesis by the regulation of the substrate.

**Conclusion.** In this work, the diverse phenotype variation of *fus3* defective strains indicated that Fus3, a global kinase, participates in several development and metabolism processes in *A. flavus*. The interaction and phosphorylation in the Fus3-MAPK pathway were confirmed by Y2H, TAP, and Western blotting, and the key phosphorylation site and the putative phosphorylation motif of Fus3 were also identified. Most importantly, the regulatory mechanism of Fus3 in AF biosynthesis was partly revealed. Briefly, Fus3 could regulate carbon metabolism relevant genes, especially AccA, then modulate acetyl-CoA and malonyl-CoA levels, and subsequently affect AF production (Fig. 7B). Additionally, several global regulators might be affected by Fus3, which are involved in stress response, sexual and asexual development, and nutrition metabolism (Fig. 7A). Our data indicate that Fus3 is a critical kinase for fungal development and secondary metabolism in *A. flavus*.

## MATERIALS AND METHODS

**Strains, media, transformation, and cultivation of the microorganisms.** The strains and plasmids used are listed in Table S4. PDA, GMM, WKM, and YES media were prepared as described previously (20, 37). Strains were cultivated on PDA, GMM, and YES media at 28°C in the dark for 7 days for detecting mycelial growth rate, conidia production, and $AFB_1$ production. For sclerotia detection, the strains were grown on WKM at 28°C in the dark for 14 days and sprayed with 75% ethanol to wash off conidia and mycelia, then sclerotia were counted on each plate.

**Construction of null-deleted, site-directed mutagenesis, and complementary strains of Fus3.** The genetic information for *ste11*, *ste7*, *fus3*, and *ste50* were downloaded from the National Center for Biotechnology Information (NCBI) database (https://www.ncbi.nlm.nih.gov/). Null-deletion strains were constructed using a homologous recombination approach as described previously (51, 55). Primers used in this study are listed in Table S5. The PCR fragments were amplified with specific primers and fused by overlap PCR (FastPfu DNA polymerase, Transgen, Beijing, China). The fusion products were transformed into protoplasts of the *A. flavus* TJES19.1 strain by using polyethylene glycol buffer (44, 52, 53, 56).

The CDS sequence of *fus3* was amplified from total cDNA and cloned into pMD18-T (D101A, TaKaRa, Dalian, China) with SmaI (JS501, Transgen), EcoRV (JE301, Transgen), and a ClonExpress II One Step Cloning Kit (C112-01, Vazyme Biotech Co., Nanjing, China). Four fragments including upstream and downstream flanking regions, the Fus3 CDS region, and PyrG were amplified and fused by overlap PCR. The complementary vector was transformed into protoplasts of Δ*fus3* to generate a complementary strain (COM). The complementary vector was also regarded as the template to generate the site-mutated sequence, and then the PCR product was eliminated from the template with DpnI (1235S,

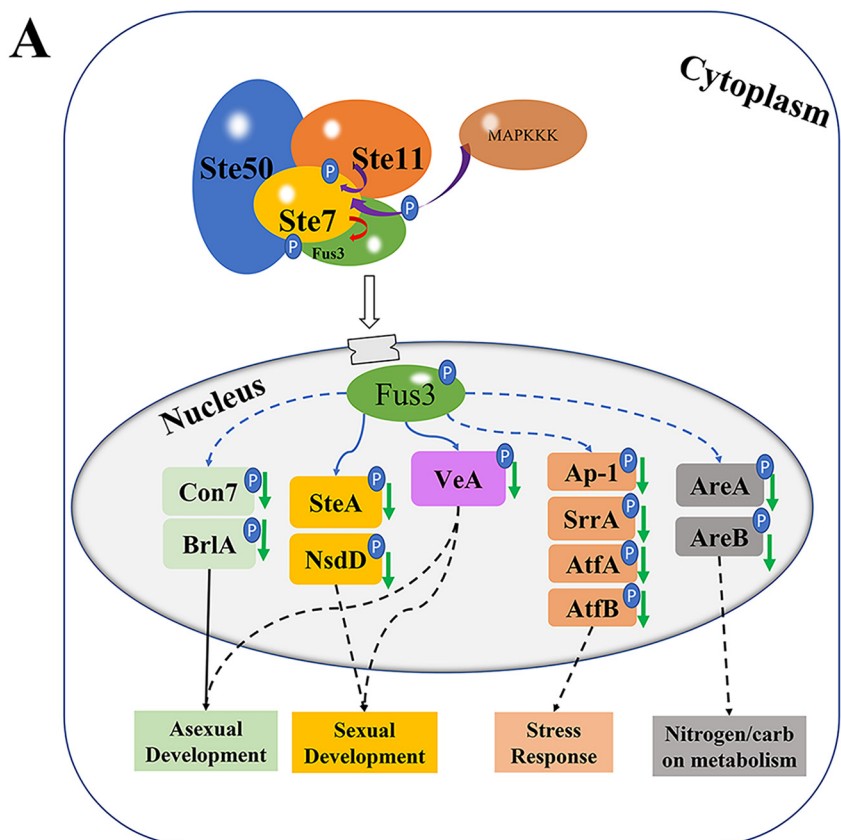

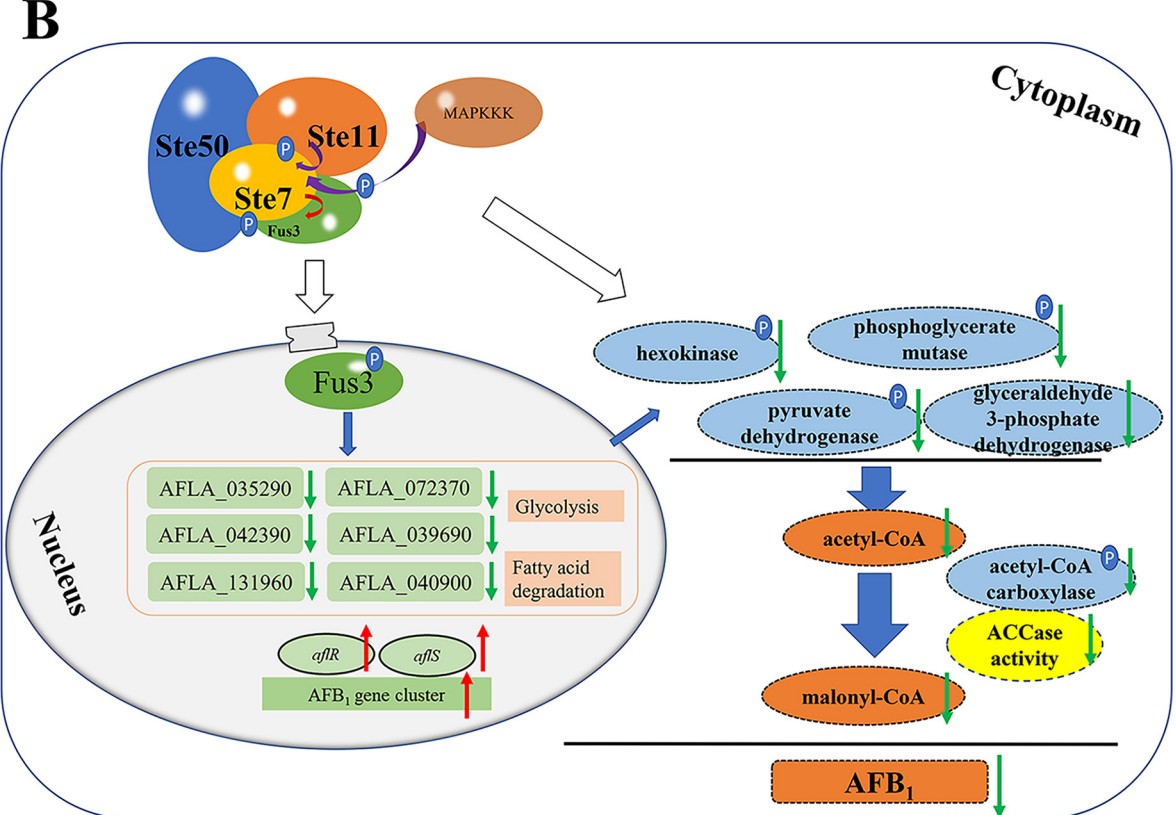

**FIG 7** Hypothetical regulatory mechanism of Fus3 on fungal development and AFs biosynthesis. The Fus3-MAPK complex consists of Ste50, Ste11, Ste7, and Fus3 proteins. (A) Fus3 could phosphorylate several the global regulators, then regulate the downstream genes

TaKaRa), and purified with a gel purification system (Magen, Beijing, China). The construction of the mutant strain with site-directed mutagenesis was similar to the null-deletion mutant construction.

**Scanning electron microscope analysis.** The mycelia were harvested after 7 days of cultivation and placed in 2.5% glutaraldehyde to fix the mycelia in the dark for 24 h. Then, the samples were treated as described in a previous report (57). Samples were visualized using a Hitachi SU8010 (Hitachi Science System, Ibaraki, Japan), and images were taken at 1,000× magnification (57, 58).

**The extraction and detection of $AFB_1$ production.** After 7 days of cultivation, $AFB_1$ levels were determined according to Liang et al. with minor modifications (59). The $AFB_1$ was extracted with acetonitrile:water (84:16) and purified using a ToxinFast immunoaffinity column (Huaan Magnech Biotech, Beijing, China). $AFB_1$ was quantified using an HPLC-FLD system with a fluorescence detector (Agilent 1220 Infinity II System, Santa Clara, USA) and a post-column derivation system (Huaan Magnech Biotech, Beijing, China), and a TC-C18 column (250 mm × 4.6 mm, 5 mm particle size; Agilent). The mobile phase was 70% methanol, and the retention time of $AFB_1$ was about 5.7 min.

**Yeast two-hybrid assay.** The CDS regions of *fus3*, *ste7*, *ste11*, and *ste11* were amplified from total cDNA, purified with a Gel Pure Kit columns I (Magen), and attached to linearized pGADT7-AD and pGBKT7-BD with a ClonExpress II One Step Cloning Kit (Vazyme Biotech Co.). The constructed plasmids were sequenced in Sangon Biotech (Shanghai, China), extracted with a TIANprep mini Plasmid Kit (Transgene, Beijing, China), then pairwise co-transfected into the Y2HGold cells by using a Yeastmaker Yeast Transformation System 2 (630439, TaKaRa). All of the selective medium and reagents, including SD/-Leu, SD/-Trp, SD/-Leu/-Trp, SD/-His/-leu/-Trp/-His, and X-α-gal, were purchased from Coolaber (Beijing, China). All primers used in this experiment are listed in Table S5.

**Tandem affinity purification.** The *sbp* and *gfp* sequences were cloned into pMD18-CZ1 with SmaI, EcoRV, and a ClonExpress II One Step Cloning Kit. Four fragments including upstream and downstream flanking regions, *sbp::fus3::gfp* region, and the *pyrG* sequence were amplified and fused by overlap PCR. The complementation vector was transformed into protoplasts of TJES 19.1 to generate the Fus3-tandem affinity purification (TAP) strain. After TAP purification, the effluent proteins were harvested and then identified with LC-MS/MS (14).

**RNA extraction, reverse transcription, and RT-qPCR analysis.** Total RNA extraction and RT-qPCR analysis were performed according to Li et al. with minor modifications (54). Mycelia were harvested after 5 days of cultivation. RNA was isolated according to the manufacturer's instructions by using an RNApure Total RNA Kit (Aidlab Biotechnologies Co., Ltd, Beijing, China). Briefly, after treatment with RNase-free DNase I, RNA samples were qualified and quantified by using a NanoDrop 2000 spectrophotometer (Thermo Fisher, Waltham, MA, USA) and Agilent 2100 Bioanalyzer (Agilent). First-strand cDNA was synthesized according to the manufacturer's instructions (Transgen, Beijing, China). SYBR green qPCR Master Mix (Transgen) and the QuantStudio 6 Flex (Applied Biosystems, Carlsbad, CA, USA) qPCR system were used to determine gene expression. The qPCR primers are listed in Table S5.

**RNA sequencing and analysis.** RNA-seq was carried out and analyzed by Novogene (Beijing, China) according to the method described by Li et al. (60, 61). In brief, after purification, the RNA libraries were constructed and then sequenced by using an Illumina Hiseq 4000 platform (San Diego, CA, USA). After filtering out low-quality reads and adaptor contamination, the clean reads were mapped to the *A. flavus* NRRL3357 genome sequence (BioProject: PRJNA13284). The gene expression changes were evaluated and the DEGs were identified with an FDR value of $\leq$ 0.05. Transcriptome raw data has been uploaded to NCBI (BioProject: PRJNA777400).

**Extraction, digestion, and LC-MS/MS analysis of proteins for phosphoproteomic analysis.** The phosphoproteomics analysis was performed and analyzed by HTHealth Technology Co. Ltd. (Beijing, China). The extraction and digestion of intracellular proteins for phosphoproteomics were performed according to a method described previously (62). Mycelia were collected, frozen with liquid nitrogen, and ground into powder. Approximately 30-mg samples were treated with lysis solution (Thermo Fisher, San Jose, CA, USA). The lysate (2 mg) was purified by using a 3 kDa ultrafiltration column (Centriprep YM-3; Millipore, Billerica, MA, USA) and precipitated by adding trichloroacetic acid with a final concentration of 10%. After an ice bath, centrifugation, and washing with cold acetone, the pellets were reduced with tris-phosphine, alkylated with iodoacetamide, and digested with trypsin (Promega, Madison, WI, USA). Phosphopeptides were enriched using a Pierce TiO2 Phosphopeptide Enrichment and Clean-up kit (Thermo Fisher).

LC-MS/MS analysis was performed as described in Ribeiro et al. (62). Enriched phosphopeptides were resuspended in 10 $\mu$L acetonitrile/$H_2O$/fatty acid (5/95/0.1) and subjected to online reverse-phase nanoLC-MS/MS analysis with 50% sample loading using an Easy nLC1000, coupled to a Q-Exactive mass spectrometer (Thermo Fisher). FTMS1 spectra were performed with regular parameters. For the ITMS2 spectra, the parameters were set as rapid scan rate, CID NCE 25, 1.6 *m/z* isolation window, AGC target 1E4, and MIT of 50 ms. The precursors were selected for a 3-s cycle, interrogated with an assigned monoisotopic *m/z*, and filtered using a 60-s dynamic exclusion window.

**Data analysis for phosphoproteomics.** Raw MS data were analyzed using MaxQuant software

**FIG 7** Legend (Continued)

expressions, and subsequently participate in diverse biological processes. (B) Fus3 could regulate several genes in glycolysis and fatty acid degradation at transcriptional level and phosphorylation level, affect the enzymes activities and the levels of acetyl-CoA and malonyl-CoA, subsequently positively modulate the $AFB_1$ production. The character P stands for the phosphorylation group, and the certified and uncertified regulatory pathways were represented with solid and dashed lines, respectively. Upregulation and downregulation in Δ*fus3* were represented with red and green arrows, respectively.

v.1.5.0.30 (25). The MS/MS spectra were mapped to the *A. flavus* Uniprot FASTA database using the following parameters: cysteine carbamidomethylation as a fixed modification and oxidation of methionine, N-acetylation of protein, and/or phosphorylation of Ser, Thr, and Tyr residues (STY) as variable modifications. The maximum peptide and site FDR were specified as 0.01. Student's *t* test with $P < 0.05$ was used to verify the statistical differences. The phosphoproteome data were submitted to iProX database with ID: IPXO003678000.

**Western blotting.** Total proteins in strains were denatured in protein-loading buffer and separated with 12% sodium dodecyl sulfate-polyacrylamide gelelectrophoresis (SDS-PAGE) (GenScript, Nanjing, China). After electrophoresis, proteins were transferred to polyvinylidene fluoride (PVDF) membranes and hybridized with the anti-phospho-site-specific Phospho-p38 MAPK antibodies (1:1,000, 4511, Cell Signaling Technology, USA) followed by incubation with a peroxidase-conjugated secondary antibody (1:5000, Biyuntian, Beijing, China). Membranes were washed three times with tris buffered saline tween (TBST), and proteins were detected using an eECL Western Blot Kit (Cowin-Biotech, Beijing, China). The results of the Western blotting were recorded with ImageQuant TL (GE Healthcare, Danderyd, Sweden).

**Detection of acetyl-coenzyme A and malonyl-coenzyme A.** The CoA-esters were extracted as described previously with minor modifications (63). The strains were inoculated in YES broth at 28°C and 100 rpm rotation, and mycelia were harvested after 72 h of cultivation. Mycelia (1 g) were lysed with 1 mL 6% perchloric acid and added 0.5 mL 3 M potassium carbonate to neutralize the reaction system. The extractions of CoA-ester were obtained after centrifugation. The detections of acetyl-CoA and malonyl-CoA were performed according to the instructions of the acetyl-CoA (MM-62538O1, mlbio, Shanghai, China) and malonyl-CoA enzyme-linked immunoassay kits (MB-10579B, mlbio), and quantified with a microplate reader (Fluoroskan Ascent FL, Thermo Fisher).

**Detection of the ACCase quantity and ACCase activity.** ACCase extraction was conducted according to Feria et al. with some modifications (63). The strains were inoculated in YES broth at 28°C and 100 rpm rotation, and mycelia were harvested after 72 h. Fresh mycelia (300 mg) were ground with liquid nitrogen, and placed into 1 mL ACCase extraction buffer (100 mM Tris-HCl, pH 8.5, 300 mM glycerol, 5 mM dithiothreitol (DTT), 2 mmol/L ethylene diamine tetraacetic acid (EDTA), 0.5 mM phenylmethylsulfonyl fluoride (PMSF), 0.1% (vol/vol) TritonX-100). The mixtures were incubated at 4°C for 2 h and then centrifuged at 4,000 g for 30 min, followed by centrifugation at 13,000 g for 30 min. The supernatant was transferred into 50% ammonium sulfate buffer, stirred for 1 h at 4°C, and then centrifuged at 13,000 g for 1 h. The precipitates were resuspended in extraction buffer.

ACCase quantity and activity were detected with an ACCase quantity assay kit (ml5669873, mlbio) and an ACCase activity assay kit (ml863669, mlbio), respectively. The OD values were detected by using a microplate reader (Fluoroskan Ascent FL, Thermo Fisher).

**Data availability.** The transcription data for this study were upload to NCBI BioProject: PRJNA777400, which can be found in https://www.ncbi.nlm.nih.gov/bioproject/PRJNA777400/; The phosphoproteome data were submitted to iProX database, which can be found in https://www.iprox.cn/page/PSV023.html;?url=1641433214564p7GZ, Password: Q3NV.

## SUPPLEMENTAL MATERIAL

Supplemental material is available online only.
**SUPPLEMENTAL FILE 1**, XLSX file, 3.8 MB.
**SUPPLEMENTAL FILE 2**, XLSX file, 0.3 MB.
**SUPPLEMENTAL FILE 3**, PDF file, 0.6 MB.

## ACKNOWLEDGMENTS

This research was funded by National Natural Science Foundation of China (31972179, 32001813), the National Key R&D Program of China (2016YFD0400105), the National Peanut Industrial Technology System (CARS-13), of Agricultural Science and Technology Innovation Program (CAAS-ASTIP-2020-IFST-03), and funded by Qingdao Science and Technology Benefit the People Demonstration and Guidance Special Project (21-1-4-NY-4-NSH).

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
