## [Reviewer comments · Microbiology Spectrum]

Microbiology Spectrum

Fus3, as a critical kinase in MAPK cascade, regulates aflatoxin biosynthesis by controlling the substrate supply in *Aspergillus flavus*, rather than the modulation of aflatoxin biosynthetic genes

Longxue Ma, Xu Li, Fuguo Xing, Junning Ma, Xiaoyun Ma, and Yiran Jiang

Corresponding Author(s): Fuguo Xing, Institute of Food Science and Technology, Chinese Academy of Agricultural Sciences

Review Timeline:

Submission Date:	August 16, 2021
Editorial Decision:	October 6, 2021
Revision Received:	December 8, 2021
Editorial Decision:	December 14, 2021
Revision Received:	January 6, 2022
Accepted:	January 7, 2022

Editor: Christina Cuomo

Reviewer(s): Disclosure of reviewer identity is with reference to reviewer comments included in decision letter(s). The following individuals involved in review of your submission have agreed to reveal their identity: Hee-Soo Park (Reviewer #2)

Transaction Report:

DOI: <https://doi.org/10.1128/Spectrum.01269-21>

October 6, 2021

Prof. Fuguo Xing
Institute of Food Science and Technology, Chinese Academy of Agricultural Sciences
Beijing
China

Re: Spectrum01269-21 (Fus3, as a critical kinase in MAPK cascade, regulates aflatoxin biosynthesis by controlling the substrate supply in *Aspergillus flavus*, rather than the regulation of cluster genes)

Dear Prof. Fuguo Xing:

Thank you for submitting your manuscript to Microbiology Spectrum. Two reviewers have provided feedback that I would like you to address in a revision. Both reviewers have highlighted additional analysis needed to support your conclusions; comments from reviewers are below and also please see the attached file for the full comments from reviewer 1. Please also ensure that a data availability statement is included that describes how to access the transcriptome and phosphoproteome data in an appropriate public repository such as NCBI (see <https://journals.asm.org/open-data-policy>).

When submitting the revised version of your paper, please provide (1) point-by-point responses to the issues raised by the reviewers as file type "Response to Reviewers," not in your cover letter, and (2) a PDF file that indicates the changes from the original submission (by highlighting or underlining the changes) as file type "Marked Up Manuscript - For Review Only". Please use this link to submit your revised manuscript - we strongly recommend that you submit your paper within the next 60 days or reach out to me. Detailed information on submitting your revised paper are below.

Link Not Available

Sincerely,

Christina Cuomo

Journals Department
Reviewer comments:

Reviewer #1 (Comments for the Author):

The mycology and transcriptome and phosphoproteome methodological were well carried out.

Reviewer #2 (Comments for the Author):

In this manuscript, the authors examined the roles of Fus3 in *Aspergillus flavus*. Fus3 is a key component in MAPK pathway and play a various role in fungal development and mycotoxin productions. To examine the role of Fus3, the authors carried out transcriptomic and phosphoproteomic analyses. Although the authors present lots of results, these results are difficult to support their conclusions. If the authors analyze these results carefully, it is thought that meaningful conclusions will be presented.

1. The results of protein interacting experiments do not fully support the results presented by the authors. In vivo protein interacting experiment such as IP assay should be needed.
2. It's not clear why the authors combine the results of phosphoproteomic and transcriptomic analyses. They selected 66 proteins which decreased phosphorylation and mRNA expression, but its' not reasonable.
3. The authors generated various phospho-mutant strains. However, it is difficult to make conclusions based on the results obtained from this experiment.
4. The authors checked mRNA expression of various genes related to development, but it's not clear what conclusions can be drawn from these results.
5. Fus3 can affect phosphorylation of TFs directly or indirectly. However, the current results do not provide any important information about the direct targets of Fus3.

Staff Comments:

Preparing Revision Guidelines

Please return the manuscript within 60 days; if you cannot complete the modification within this time period, please contact me. If you do not wish to modify the manuscript and prefer to submit it to another journal, please notify me of your decision immediately so that the manuscript may be formally withdrawn from consideration by Microbiology Spectrum.

In this study, the authors determined MAPK pathway genes function, confirmed that Ste50-Ste11-Ste7-Fus3 protein interactions and phosphorylations, explored the possible phosphorylation motifs and potential targets of the terminal kinase Fus3, and illustrated Fus3 responding to diverse environment stresses in *Aspergillus*. They revealed the mechanism of Fus3 positive regulation on AFs biosynthesis. $\Delta fus3$ mutant showed down-regulation of AFs production, but up-regulation of AFs cluster genes. The substrate of AFs, acetyl-CoA and malonyl-CoA, were significantly decreased in *fus3* null-deletion and site-mutagenesis strains, and the genes involved in acetyl-CoA and malonyl-CoA biosynthesis, were significantly down-regulated at transcriptional or phosphorylational levels. This paper is a interesting topic, and the experiments were well designed. It is meaningful for readers in related fields. So my suggestion is acceptable after minor revision. Other comments were followed.

- 1) All the transcriptome and phosphoproteome data should be submitted to NCBI.
- 2) In Fig.4, Four mutants (T182A, T182D, T184A, T184D) showed almost the same phenotype, why? Please explain.
- 3) There are many important MAPK in *Aspergillus flavus*, why the author chose the Fus3, please explain in Introduction part.
- 4) Fig.3 showed the transcriptome and phosphoproteome results, but transcriptome only give small partial. Please provide more information on transcriptome.
- 5) Line 217, "But the expressions of AFs biosynthetic genes were no obvious decrease in *fus3* deletion" The author should make clear in which level, transcription or translation?
- 6) The English writing should be improved by an English native speaker.
- 7) Fig.8 is not clear, which genes correspond to which phenotype, should be clear and right.

Reviewer #1:

The mycology and transcriptome and phosphoproteome methodological were well carried out. In this study, the authors determined MAPK pathway gene function, confirmed that Ste50-Ste11-Ste7-Fus3 protein interactions and phosphorylations, explored the possible phosphorylation motifs and potential targets of the terminal kinase Fus3, and illustrated Fus3 responding to diverse environment stresses in *Aspergillus*. They revealed the mechanism of Fus3 positive regulation on AFs biosynthesis. $\Delta fus3$ mutant showed down-regulation of AFs production, but up-regulation of AFs cluster genes. The substrate of AFs, acetyl-CoA and malonyl-CoA were significantly decreased in *fus3* null-deletion and site-mutagenesis strains, and the genes involved in acetyl-CoA and malonyl-CoA biosynthesis, were significantly down-regulated at transcriptional or phosphorylation levels. This paper is an interesting topic, and the experiments were well designed. It is meaningful for readers in related fields. So, my suggestion is acceptable after minor revision.

Response:

Thank you for the kind summary and review on our manuscript. And in this revised edition, the context was optimized, more information, including Fus3-TAP and ACCase analyses, were added, the language was revised by the native speaker, and some minor mistakes were corrected. The revised manuscript was more logical, and more information was supplied about the regulation of Fus3 on AFs biosynthesis. Thanks again for your serious and careful review on our manuscript. It is truly helpful to improve our manuscript.

Other comments were followed.

1) All the transcriptome and phosphoproteome data should be submitted to NCBI.

Response:

Thanks for your comment. The transcriptome data was submitted to NCBI and the transcriptome data ID is PRJNA777400 (**Line 531**). In the newest edition, the phosphoroteome data was submitted to iProX database, and the ID is IPXO003678000 with the share link (<https://www.iprox.cn/page/SSV024.html?url=1638843223734VUiQ>, Psssword: qZIU) (**Line 562-563**).

2) In Fig.4, four mutants (T182A, T182D, T184A, T184D) showed almost the same phenotype, why? Please explain.

Response:

Thanks for your comment. In this study, the two phosphorylation sites, T182 and Y184, were identified by the phosphoproteome, which is accordant with Yang et al. reported. By generating the site mutagenesis strains, we noticed that the phenotypes of *fus3*^{T182A} and *fus3*^{Y184A} strains were impaired in conidia production, sclerotia formation and AFs biosynthesis (Fig. 6). Because T182 and Y184 are the critical phosphorylation sites of Fus3, *fus3*^{T182A} and *fus3*^{Y184A} abolished the phosphorylation function of Fus3, and their phenotypes are similar with the *fus3* null-deleted strain. But unexpectedly, *fus3*^{T182D} and *fus3*^{Y184D} did not recover the WT phenotypes, but still consisted with the $\Delta fus3$ phenotypes (Fig. 6).

A lot of phosphorylation studies found that the replacements of Thr/Tyr residues with Asp are not always perfectly mimic the constitutively phosphorylation phenotypes. The possible reason is that the variation of amino acid leads to the changes in protein high-level structure and proteins interaction, which results in the defective protein functions and the mutant phenotypes. In our research, we also noticed that the interactions of Fus3^{T182A}, Fus3^{T182D}, Fus3^{Y184A} and Fus3^{Y184D} with MAPK proteins. As shown, Fus3^{T182D} could not interact with the other MAPK modules, suggested that the replacement of threonine with aspartic acid obviously affected the interactions and the phosphorylation signal transduction, which abolished the normally function of Fus3.

Yang G, Cao X, Ma G, Qin L, Wu Y, Lin J, Ye P, Yuan J, Wang S. 2020. MAPK pathway-related tyrosine phosphatases regulate development, secondary metabolism and pathogenicity in fungus *Aspergillus flavus*. *Environ Microbiol* **22**:5232-5247.

3) There are many important MAPK in *Aspergillus flavus*, why the author chose the Fus3, please explain in Introduction part.

Response:

Thanks for your suggestion. Diverse MAPK signaling pathways were recognized in *A. flavus*, such as SakA-MAPK, Slt2-MAPK, and Fus3-MAPK. Fus3-MAPK cascade was paid more

attention due to its significant effect on aflatoxin biosynthesis (Ren et al., 2016) (Line 82-85). Fus3, as the terminal kinase of MAPK pathway, could directly affect the expressions and the functions of several regulators, for example VeA, SteA (Bayram et al., 2012) (Line 90-91). More important, AFs, the most critical secondary metabolism of *A. flavus*, are positively regulated by Fus3 (Frawley et al., 2019; Yang et al., 2020) (Line 104-106). And our previous research also found the inhibition of eugenol on AFs is closely relevant with Fus3 (Lv et al., 2018). But the specific regulatory mechanism of Fus3 on AFs biosynthesis is still unclear. Therefore, we focused on the function and regulation of Fus3 in this study.

Ren S, Yang M, Li Y, Zhang F, Chen Z, Zhang J, Yang G, Yue Y, Li S, Ge F. 2016. Global phosphoproteomic analysis reveals the involvement of phosphorylation in aflatoxins biosynthesis in the pathogenic fungus *Aspergillus flavus*. *Scientific Reports* **6**:34078.

Yang G, Cao X, Ma G, Qin L, Wu Y, Lin J, Ye P, Yuan J, Wang S. 2020. MAPK pathway-related tyrosine phosphatases regulate development, secondary metabolism and pathogenicity in fungus *Aspergillus flavus*. *Environ Microbiol* **22**:5232-5247.

Bayram O, Bayram OS, Ahmed YL, Maruyama J, Valerius O, Rizzoli SO, Ficner R, Irniger S, Braus GH. 2012. The *Aspergillus nidulans* MAPK module AnSte11-Ste50-Ste7-Fus3 controls development and secondary metabolism. *PLoS Genet* **8**:e1002816.

Lv C, Wang P, Ma L, Zheng M, Liu Y, Xing F. 2018. Large-scale comparative analysis of eugenol-induced/repressed genes expression in *Aspergillus flavus* using RNA-seq. *Frontiers in microbiology*, **9**, 1116.

4) Fig.3 showed the transcriptome and phosphoproteome results, but transcriptome only give small partial. Please provide more information on transcriptome.

Response:

Thanks for your comment. Firstly, Fus3 as a phosphokinase, would directly regulate the phosphorylation levels of the downstream genes. The variations in transcriptional level in $\Delta fus3$ should be resulted from the indirect regulation of Fus3. In order to find the potential target of Fus3, we paid more attention in phosphoproteome analysis. But we still discovered a lot of useful information from the transcriptome data. The transcriptional level changes of AFs related genes

and developmental genes in $\Delta fus3$ partly illustrated the function and regulation of Fus3. And the transcriptome data supported our conclusion (Fig. 5).

So, as your suggestion, the transcriptome results were added in the revised manuscript as the supplementary materials (Table S1).

5) Line 217, “But the expressions of AFs biosynthetic genes were no obvious decrease in *fus3* deletion” The author should make clear in which level, transcription or translation?

Response:

Sorry for the writing mistake. The sentence was corrected as “It is paradoxical that AFs cluster genes were up-regulated at transcriptional level, but AFs production was decreased in $\Delta fus3$ ” in Line 256.

6) The English writing should be improved by an English native speaker.

Response:

Thank you for your suggestion. We have carefully checked the context and improved our language by the native speaker.

7) Fig.8 is not clear, which genes correspond to which phenotype, should be clear and right

Response:

Thanks for your comment. In the revised manuscript, the figure was divided into two parts, the regulation of Fus3 on fungal development and the regulation on AFs biosynthesis. And the directions of arrows are more explicit and certain, which is more easily to read and understand.

Reviewer #2:

In this manuscript, the authors examined the roles of Fus3 in *Aspergillus flavus*. Fus3 is a key component in MAPK pathway and play a various role in fungal development and mycotoxin productions. To examine the role of Fus3, the authors carried out transcriptomic and phosphoproteomic analyses. Although the authors present lots of results, these results are difficult to support their conclusions. If the authors analyze these results carefully, it is thought that meaningful conclusions will be presented.

Response:

Thank you for your kind summary and review on our manuscript. In the revised manuscript, we changed some imprecise statements and conclusions, paid more attention to the regulation of Fus3 on AFs biosynthesis, and added more experiments to support our conclusions. The mainly conclusion in the revised manuscript is “Fus3 could regulate carbon metabolism genes’ expressions at transcriptional and phosphorylation level, especially AccA, then control the levels of AFs biosynthetic precursors, acetyl-CoA and malonyl-CoA, and subsequently affect the AFs production” (Line 423-426). And the consecution and logical relationship of the context were adjusted. The revised manuscript is more credible and logical than before.

To be specific, we constructed the SBP::Fus3::GFP strain, and performed the Fus3-TAP (tandem affinity purification) analysis, which supplied the reliable evidences of Fus3 interactions. Fus3, Ste7, Ste11, Ste50 and AccA were identified by Fus3-TAP analysis (Table S3). And the analyses of ACCase quantity and ACCase indicated that the defective Fus3 significantly inhibit the ACCase activity, which confirmed the regulations of Fus3 on AccA, malonyl-CoA and AFs production.

Thanks again for your serious and careful review on our manuscript. Your suggestions are truly helpful to improve our manuscript.

1. The results of protein interacting experiments do not fully support the results presented by the authors. *In vivo* protein interacting experiment such as IP assay should be needed.

Response:

Thanks for your suggestion. To better investigating the Fus3 interactions, we constructed the

SBP::Fus3::GFP strain, with SBP and GFP tags for twice protein purification. The Fus3-TAP results were shown in Table S3. Total of 862 proteins were found by LC-MS/MS. Ste50, Ste7 and Ste11, recognized in the list, which confirmed the direct interactions between Fus3 and the other MAPK modules (**Line 139-140**). AccA was also identified as the potential target of Fus3, which supported that Fus3 could regulate AFs biosynthesis depending on the modulation of AFs substrate supplement (**Line 291-293**). The TAP analysis is helpful to support our conclusion.

2. It's not clear why the authors combine the results of phosphoproteomic and transcriptomic analyses. They selected 66 proteins which decreased phosphorylation and mRNA expression, but its' not reasonable.

Response:

Fus3, as a phosphokinase, could positively regulated the phosphorylation level of downstream targets. The transcriptional variations must be result from the indirect regulations of Fus3. So, we focused on the down-regulated phosphorylation level genes in *fus3* deletion strain.

Phosphoproteome analysis were performed using the method of Pierce TiO₂ Phosphopeptide Enrichment. The genes' expression must disturb the result of phosphopeptide enrichment. So, to reduce this disturbance, we searched the Fus3 potential targets from the proteins, which the phosphorylation levels were decreased, but the transcriptional levels were not significantly decreased in the mean while. The similar analysis method was also used in the previous study (Mattos et al., 2020).

And in the revised manuscript, the phrase was changed as “To reduce the interferences from the decrease in mRNA level, total of 1033 down-regulated phosphorylation proteins in $\Delta fus3$ were regarded as the potential Fus3 targets, excluding 66 proteins with significant down-regulation at transcriptional level” in **Line 168-170**.

Mattos EC, Silva LP, Valero C, de Castro PA. 2020. The *Aspergillus fumigatus* phosphoproteome reveals roles of high-osmolarity glycerol mitogen-activated protein kinases in promoting cell wall damage and caspofungin tolerance. *mBio* **11**(1):e02962-19. doi:10.1128/mBio.02962-19

3. The authors generated various phospho-mutant strains. However, it is difficult to make conclusions based on the results obtained from this experiment.

Response:

In this study, we generated the phosphor-site mutagenesis, and sever experiments were performed to investigate the variation of Fus3. We believe that these investigations about the site mutagenesis strains supported our conclusion.

Firstly, the phenotypes of the phosphate-site strains were similar with $\Delta fus3$, demonstrating that these phosphate-site mutagenesis directly abolish the normal function of Fus3. The variations of the critical phosphate-site directly disturb the phosphate signaling transmission.

Secondly, the ACCase activity of the phosphate-site mutagenesis strains were significantly decreased, which indicated the phosphorylation function of Fus3 is critical for ACCase activity. The previous study reported the phosphorylation level of AccA affects its activity (Choi et al., 2014; Jia et al., 2017). So, the phosphate-site mutagenesis strains confirmed the regulation of Fus3 on AccA at phosphorylation level. Furtherly, combined the phosphoproteome, TAP, and the phosphorylation motif analyses, we speculated that AccA might be a direct target of Fus3.

Taken, the investigations of phosphate-site mutagenesis strains were useful to support our conclusion.

Wei J, Zhang Y, Yu TY, Sadre-Bazzaz K, Rudolph MJ, Amodeo GA, Symington LS, Walz T, Tong L. 2017. A unified molecular mechanism for the regulation of acetyl-coa carboxylase by phosphorylation. *Cell Discovery*. **3**, 16055. doi:10.1038/celldisc.2016.44

Choi JW, Silva N. 2014. Improving polyketide and fatty acid synthesis by engineering of the yeast acetyl-CoA carboxylase. *Journal of Biotechnology*, **187**, 56-59. doi:10.1016/j.jbiotec.2014.07.430

4. The authors checked mRNA expression of various genes related to development, but it's not clear what conclusions can be drawn from these results.

Response:

Thanks for your comment. We found that defect of *fus3* obviously leaded to the impairment of fungal development, such as conidia production and sclerotia formation. And based on the

transcriptome data, we notice a lot of developmental genes, such as *con6*, *con10*, *rodA*, *brlA* and *nsdD*, showed significantly decrease in $\Delta fus3$. We believe that these reductions are due to the phosphorylation level decrease of the upstream regulators, such as VeA, SteA, Con7, NsdD. The phosphorylation level of these regulator could affect their transcriptional regulating activity (Odenbach et al., 2007). Therefore, the transcriptional expressions of critical genes related development are accordant with the phenotype variations. In this study, we mainly focused on the regulation of Fus3 on AFs regulation. So, our research is lack of more solid information about the regulation of Fus3 on fungal. We only described a possible regulatory pathway of Fus3 on fungal development. And in the revised manuscript, we deleted the overstate expressions and the uncertain conclusions. The revised manuscript is more rigorous than before. Thanks for your suggestions again.

Odenbach D, Breth B, Thines E, Weber RW, Anke H, Foster AJ. 2007. The transcription factor Con7p is a central regulator of infection-related morphogenesis in the rice blast fungus *Magnaporthe grisea*. *Mol Microbiol* **64**:293-307. doi:10.1111/j.1365-2958.2007.05643.x

5. Fus3 can affect phosphorylation of TFs directly or indirectly. However, the current results do not provide any important information about the direct targets of Fus3.

Response:

Thank you for your comments. Based on our phosphoproteome data, we noticed that several TFs showed the down-regulated phosphorylation levels in $\Delta fus3$, indicating these TFs might be directly or indirectly regulated by Fus3 at phosphorylation levels. At first, we summarized the down-regulated phosphorylation TFs to investigate the Fus3 regulations and functions.

Several TFs have been regarded as the direct target of Fus3, such as VeA, Con7, and SteA (Odenbach et al., 2007; Bayram et al., 2012; Katayama T et al., 2021). Deletion of *fus3* leads to the down-regulated phosphorylation level, and furtherly results in the decreased transcription of downstream genes and the defect development phenotype. And according to the potential targets, we predicted that the phosphorylation motif of Fus3 might be the [RxxSP] in *A. flavus*.

Importantly, we also identified that AccA, not as a TF, might be potential target of Fus3. The phosphorylation level of AccA was down-regulated in $\Delta fus3$, the direct interaction of AccA and

Fus3 were noticed by Fus3-TAP analysis, and the phosphorylation motif [RxxSP] was also identified in AccA. All information suggested that AccA could be a direct target of Fus3. Furtherly, the phosphorylation level of AccA affects ACCase activity (Jia et al., 2017). In our research, the ACCase activities in Fus3 defective strains were significantly decreased, while the ACCase activity of *fus3*^{T182D} was partly recovered (Fig. 6G). So, in our research, Fus3 could directly regulate the phosphorylation of AccA, and then affect the ACCase activity and malonyl-CoA level, subsequently positively modulate the AFs production. This is the mainly conclusion in our manuscript.

In addition, we also noticed the down-regulated phosphorylation level of other TFs, such as AtfA, AtfB, AP-1. Previous reports showed that phosphorylation levels are critical for their transcriptional activated functions (Choi, 2014; Sanchez-Mir et al., 2020). But in our research, Fus3 regulated AFs independent on AFs cluster genes and these regulators. So, we did not perform the further investigations, and we moved these TFs information to the supplementary materials in the revised manuscript (Table S2). We might discover more Fus3 targets and study more Fus3 regulations in future research.

Wei J, Zhang Y, Yu TY, Sadre-Bazzaz K, Rudolph MJ, Amodeo GA, Symington LS, Walz T, Tong L. 2017. A unified molecular mechanism for the regulation of acetyl-coa carboxylase by phosphorylation. *Cell Discovery*. **3**, 16055. doi:10.1038/celldisc.2016.44

Sánchez-Mir L, Fraile R, Ayté J, Hidalgo E. 2020. Phosphorylation of the Transcription factor Atf1 at multiple sites by the MAP kinase sty1 controls homologous recombination and transcription. *J Mol Biol*. **432**(19):5430-5446. doi:10.1016/j.jmb.2020.08.004

Choi WJ. 2014. The heterochromatin-1 phosphorylation contributes to TPA-induced AP-1 expression. *Biomol Ther (Seoul)*. **22**(4):308-313. doi:10.4062/biomolther.2014.057

December 14, 2021

Prof. Fuguo Xing
Institute of Food Science and Technology, Chinese Academy of Agricultural Sciences
Beijing
China

Re: Spectrum01269-21R1 (Fus3, as a critical kinase in MAPK cascade, regulates aflatoxin biosynthesis by controlling the substrate supply in *Aspergillus flavus*, rather than the modulation of aflatoxin biosynthetic genes)

Dear Prof. Fuguo Xing:

Thank you for submitting your revision and addressing the reviewer comments. While we are willing to consider a revised version of this paper at Spectrum, our editorial assessment is that the writing still needs substantial improvement. As I am certain you can appreciate, this is also in your best interest so that issues with incorrect wording do not discourage others from reading your work. One option is that you could ask a colleague who is a native English speaker to provide you comprehensive feedback on the writing. We would alternatively suggest that you are also welcome to use one of the services recommended by ASM:

<https://journals.asm.org/content/language-editing-services>

As these revisions are quite minor, I expect that you should be able to turn in the revised paper in less than 30 days, if not sooner.

When submitting the revised version of your paper, please provide (1) point-by-point responses to the issues I raised in your cover letter, and (2) a PDF file that indicates the changes from the original submission (by highlighting or underlining the changes) as file type "Marked Up Manuscript - For Review Only". Please use this link to submit your revised manuscript. Detailed instructions on submitting your revised paper are below.

Link Not Available

Sincerely,

Christina Cuomo

Reviewer comments:

Preparing Revision Guidelines

- point-by-point responses to the issues I raised in your cover letter
- Upload a compare copy of the manuscript (without figures) as a "Marked-Up Manuscript" file.
- Each figure must be uploaded as a separate file, and any multipanel figures must be assembled into one file.
- Manuscript: A .DOC version of the revised manuscript
- Figures: Editable, high-resolution, individual figure files are required at revision, TIFF or EPS files are preferred

Please return the manuscript within 60 days; if you cannot complete the modification within this time period, please contact me. If you do not wish to modify the manuscript and prefer to submit it to another journal, please notify me of your decision immediately so that the manuscript may be formally withdrawn from consideration by Microbiology Spectrum.

Reviewer #1:

The mycology and transcriptome and phosphoproteome methodological were well carried out. In this study, the authors determined MAPK pathway gene function, confirmed that Ste50-Ste11-Ste7-Fus3 protein interactions and phosphorylations, explored the possible phosphorylation motifs and potential targets of the terminal kinase Fus3, and illustrated Fus3 responding to diverse environment stresses in *Aspergillus*. They revealed the mechanism of Fus3 positive regulation on AFs biosynthesis. $\Delta fus3$ mutant showed down-regulation of AFs production, but up-regulation of AFs cluster genes. The substrate of AFs, acetyl-CoA and malonyl-CoA were significantly decreased in *fus3* null-deletion and site-mutagenesis strains, and the genes involved in acetyl-CoA and malonyl-CoA biosynthesis, were significantly down-regulated at transcriptional or phosphorylation levels. This paper is an interesting topic, and the experiments were well designed. It is meaningful for readers in related fields. So, my suggestion is acceptable after minor revision.

Response:

Thank you for the kind summary and review on our manuscript. And in this revised edition, the context was optimized, more information, including Fus3-TAP and ACCase analyses, were added, the language was revised by the native speaker, and some minor mistakes were corrected. The revised manuscript was more logical, and more information was supplied about the regulation of Fus3 on AFs biosynthesis. Thanks again for your serious and careful review on our manuscript. It is truly helpful to improve our manuscript.

Other comments were followed.

1) All the transcriptome and phosphoproteome data should be submitted to NCBI.

Response:

Thanks for your comment. The transcriptome data was submitted to NCBI and the transcriptome data ID is PRJNA777400 (**Line 523**). In the newest edition, the phosphoroteome data was submitted to iProX database, and the ID is IPXO003678000 with the share link (<https://www.iprox.cn/page/SSV024.html?url=1638843223734VUiQ>, Psssword: qZIU) (**Line 555-556**).

2) In Fig.4, four mutants (T182A, T182D, T184A, T184D) showed almost the same phenotype, why? Please explain.

Response:

Thanks for your comment. In this study, the two phosphorylation sites, T182 and Y184, were identified by the phosphoproteome, which is accordant with Yang et al. reported. By generating the site mutagenesis strains, we noticed that the phenotypes of *fus3*^{T182A} and *fus3*^{Y184A} strains were impaired in conidia production, sclerotia formation and AFs biosynthesis (Fig. 6). Because T182 and Y184 are the critical phosphorylation sites of Fus3, *fus3*^{T182A} and *fus3*^{Y184A} abolished the phosphorylation function of Fus3, and their phenotypes are similar with the *fus3* null-deleted strain. But unexpectedly, *fus3*^{T182D} and *fus3*^{Y184D} did not recover the WT phenotypes, but still consisted with the $\Delta fus3$ phenotypes (Fig. 6).

A lot of phosphorylation studies found that the replacements of Thr/Tyr residues with Asp are not always perfectly mimic the constitutively phosphorylation phenotypes. The possible reason is that the variation of amino acid leads to the changes in protein high-level structure and proteins interaction, which results in the defective protein functions and the mutant phenotypes. In our research, we also noticed that the interactions of Fus3^{T182A}, Fus3^{T182D}, Fus3^{Y184A} and Fus3^{Y184D} with MAPK proteins. As shown, Fus3^{T182D} could not interact with the other MAPK modules, suggested that the replacement of threonine with aspartic acid obviously affected the interactions and the phosphorylation signal transduction, which abolished the normally function of Fus3.

Yang G, Cao X, Ma G, Qin L, Wu Y, Lin J, Ye P, Yuan J, Wang S. 2020. MAPK pathway-related tyrosine phosphatases regulate development, secondary metabolism and pathogenicity in fungus *Aspergillus flavus*. *Environ Microbiol* **22**:5232-5247.

3) There are many important MAPK in *Aspergillus flavus*, why the author chose the Fus3, please explain in Introduction part.

Response:

Thanks for your suggestion. Diverse MAPK signaling pathways were recognized in *A. flavus*, such as SakA-MAPK, Slt2-MAPK, and Fus3-MAPK. Fus3-MAPK cascade was paid more

attention due to its significant effect on aflatoxin biosynthesis (Ren et al., 2016) (Line 77-80). Fus3, as the terminal kinase of MAPK pathway, could directly affect the expressions and the functions of several regulators, for example VeA, SteA (Bayram et al., 2012) (Line 85-86). More important, AFs, the most critical secondary metabolism of *A. flavus*, are positively regulated by Fus3 (Frawley et al., 2019; Yang et al., 2020) (Line 99-100). And our previous research also found the inhibition of eugenol on AFs is closely relevant with Fus3 (Lv et al., 2018). But the specific regulatory mechanism of Fus3 on AFs biosynthesis is still unclear. Therefore, we focused on the function and regulation of Fus3 in this study.

Ren S, Yang M, Li Y, Zhang F, Chen Z, Zhang J, Yang G, Yue Y, Li S, Ge F. 2016. Global phosphoproteomic analysis reveals the involvement of phosphorylation in aflatoxins biosynthesis in the pathogenic fungus *Aspergillus flavus*. *Scientific Reports* **6**:34078.

Yang G, Cao X, Ma G, Qin L, Wu Y, Lin J, Ye P, Yuan J, Wang S. 2020. MAPK pathway-related tyrosine phosphatases regulate development, secondary metabolism and pathogenicity in fungus *Aspergillus flavus*. *Environ Microbiol* **22**:5232-5247.

Bayram O, Bayram OS, Ahmed YL, Maruyama J, Valerius O, Rizzoli SO, Ficner R, Irniger S, Braus GH. 2012. The *Aspergillus nidulans* MAPK module AnSte11-Ste50-Ste7-Fus3 controls development and secondary metabolism. *PLoS Genet* **8**:e1002816.

Lv C, Wang P, Ma L, Zheng M, Liu Y, Xing F. 2018. Large-scale comparative analysis of eugenol-induced/repressed genes expression in *Aspergillus flavus* using RNA-seq. *Frontiers in microbiology*, **9**, 1116.

4) Fig.3 showed the transcriptome and phosphoproteome results, but transcriptome only give small partial. Please provide more information on transcriptome.

Response:

Thanks for your comment. Firstly, Fus3 as a phosphokinase, would directly regulate the phosphorylation levels of the downstream genes. The variations in transcriptional level in $\Delta fus3$ should be resulted from the indirect regulation of Fus3. In order to find the potential target of Fus3, we paid more attention in phosphoproteome analysis. But we still discovered a lot of useful information from the transcriptome data. The transcriptional level changes of AFs related genes

and developmental genes in $\Delta fus3$ partly illustrated the function and regulation of Fus3. And the transcriptome data supported our conclusion (Fig. 5).

So, as your suggestion, the transcriptome results were added in the revised manuscript as the supplementary materials (Table S1).

5) Line 217, “But the expressions of AFs biosynthetic genes were no obvious decrease in *fus3* deletion” The author should make clear in which level, transcription or translation?

Response:

Sorry for the writing mistake. The sentence was corrected as “It is paradoxical that AFs cluster genes were up-regulated at transcriptional level, but AFs production was decreased in $\Delta fus3$ ” in Line 251.

6) The English writing should be improved by an English native speaker.

Response:

Thank you for your suggestion. We have carefully checked the context and improved our language by the native speaker.

7) Fig.8 is not clear, which genes correspond to which phenotype, should be clear and right

Response:

Thanks for your comment. In the revised manuscript, the figure was divided into two parts, the regulation of Fus3 on fungal development and the regulation on AFs biosynthesis. And the directions of arrows are more explicit and certain, which is more easily to read and understand.

Reviewer #2:

In this manuscript, the authors examined the roles of Fus3 in *Aspergillus flavus*. Fus3 is a key component in MAPK pathway and play a various role in fungal development and mycotoxin productions. To examine the role of Fus3, the authors carried out transcriptomic and phosphoproteomic analyses. Although the authors present lots of results, these results are difficult to support their conclusions. If the authors analyze these results carefully, it is thought that meaningful conclusions will be presented.

Response:

Thank you for your kind summary and review on our manuscript. In the revised manuscript, we changed some imprecise statements and conclusions, paid more attention to the regulation of Fus3 on AFs biosynthesis, and added more experiments to support our conclusions. The mainly conclusion in the revised manuscript is “Fus3 could regulate the expression of carbon metabolism genes at the transcriptional and phosphorylation level, especially AccA, then control the levels of AF biosynthetic precursors, acetyl-CoA and malonyl-CoA, and subsequently affect AF production” (Line 418-421). And the consecution and logical relationship of the context were adjusted. The revised manuscript is more credible and logical than before.

To be specific, we constructed the SBP::Fus3::GFP strain, and performed the Fus3-TAP (tandem affinity purification) analysis, which supplied the reliable evidences of Fus3 interactions. Fus3, Ste7, Ste11, Ste50 and AccA were identified by Fus3-TAP analysis (Table S3). And the analyses of ACCase quantity and ACCase indicated that the defective Fus3 significantly inhibit the ACCase activity, which confirmed the regulations of Fus3 on AccA, malonyl-CoA and AFs production.

Thanks again for your serious and careful review on our manuscript. Your suggestions are truly helpful to improve our manuscript.

1. The results of protein interacting experiments do not fully support the results presented by the authors. *In vivo* protein interacting experiment such as IP assay should be needed.

Response:

Thanks for your suggestion. To better investigating the Fus3 interactions, we constructed the

SBP::Fus3::GFP strain, with SBP and GFP tags for twice protein purification. The Fus3-TAP results were shown in Table S3. A total of 862 proteins were found by LC-MS/MS. Ste50, Ste7 and Ste11, recognized in the list, which confirmed the direct interactions between Fus3 and the other MAPK modules (**Line 133-135**). AccA was also identified as the potential target of Fus3, which supported that Fus3 could regulate AFs biosynthesis depending on the modulation of AFs substrate supplement (**Line 287-288**). The TAP analysis is helpful to support our conclusion.

2. It's not clear why the authors combine the results of phosphoproteomic and transcriptomic analyses. They selected 66 proteins which decreased phosphorylation and mRNA expression, but its' not reasonable.

Response:

Fus3, as a phosphokinase, could positively regulated the phosphorylation level of downstream targets. The transcriptional variations must be result from the indirect regulations of Fus3. So, we focused on the down-regulated phosphorylation level genes in *fus3* deletion strain.

Phosphoproteome analysis were performed using the method of Pierce TiO₂ Phosphopeptide Enrichment. The genes' expression must disturb the result of phosphopeptide enrichment. So, to reduce this disturbance, we searched the Fus3 potential targets from the proteins, which the phosphorylation levels were decreased, but the transcriptional levels were not significantly decreased in the mean while. The similar analysis method was also used in the previous study (Mattos et al., 2020).

And in the revised manuscript, the phrase was changed as “To reduce the interferences from the decrease in mRNA level, total of 1033 down-regulated phosphorylation proteins in $\Delta fus3$ were regarded as the potential Fus3 targets, excluding 66 proteins with significant down-regulation at transcriptional level” in **Line 163-166**.

Mattos EC, Silva LP, Valero C, de Castro PA. 2020. The *Aspergillus fumigatus* phosphoproteome reveals roles of high-osmolarity glycerol mitogen-activated protein kinases in promoting cell wall damage and caspofungin tolerance. *mBio* **11**(1):e02962-19. doi:10.1128/mBio.02962-19

3. The authors generated various phospho-mutant strains. However, it is difficult to make conclusions based on the results obtained from this experiment.

Response:

In this study, we generated the phosphor-site mutagenesis, and sever experiments were performed to investigate the variation of Fus3. We believe that these investigations about the site mutagenesis strains supported our conclusion.

Firstly, the phenotypes of the phosphate-site strains were similar with $\Delta fus3$, demonstrating that these phosphate-site mutagenesis directly abolish the normal function of Fus3. The variations of the critical phosphate-site directly disturb the phosphate signaling transmission.

Secondly, the ACCase activity of the phosphate-site mutagenesis strains were significantly decreased, which indicated the phosphorylation function of Fus3 is critical for ACCase activity. The previous study reported the phosphorylation level of AccA affects its activity (Choi et al., 2014; Jia et al., 2017). So, the phosphate-site mutagenesis strains confirmed the regulation of Fus3 on AccA at phosphorylation level. Furtherly, combined the phosphoproteome, TAP, and the phosphorylation motif analyses, we speculated that AccA might be a direct target of Fus3.

Taken, the investigations of phosphate-site mutagenesis strains were useful to support our conclusion.

Wei J, Zhang Y, Yu TY, Sadre-Bazzaz K, Rudolph MJ, Amodeo GA, Symington LS, Walz T, Tong L. 2017. A unified molecular mechanism for the regulation of acetyl-coa carboxylase by phosphorylation. *Cell Discovery*. **3**, 16055. doi:10.1038/celldisc.2016.44

Choi JW, Silva N. 2014. Improving polyketide and fatty acid synthesis by engineering of the yeast acetyl-CoA carboxylase. *Journal of Biotechnology*, **187**, 56-59. doi:10.1016/j.jbiotec.2014.07.430

4. The authors checked mRNA expression of various genes related to development, but it's not clear what conclusions can be drawn from these results.

Response:

Thanks for your comment. We found that defect of *fus3* obviously leaded to the impairment of fungal development, such as conidia production and sclerotia formation. And based on the

transcriptome data, we notice a lot of developmental genes, such as *con6*, *con10*, *rodA*, *brlA* and *nsdD*, showed significantly decrease in $\Delta fus3$. We believe that these reductions are due to the phosphorylation level decrease of the upstream regulators, such as VeA, SteA, Con7, NsdD. The phosphorylation level of these regulator could affect their transcriptional regulating activity (Odenbach et al., 2007). Therefore, the transcriptional expressions of critical genes related development are accordant with the phenotype variations. In this study, we mainly focused on the regulation of Fus3 on AFs regulation. So, our research is lack of more solid information about the regulation of Fus3 on fungal. We only described a possible regulatory pathway of Fus3 on fungal development. And in the revised manuscript, we deleted the overstate expressions and the uncertain conclusions. The revised manuscript is more rigorous than before. Thanks for your suggestions again.

Odenbach D, Breth B, Thines E, Weber RW, Anke H, Foster AJ. 2007. The transcription factor Con7p is a central regulator of infection-related morphogenesis in the rice blast fungus *Magnaporthe grisea*. *Mol Microbiol* **64**:293-307. doi:10.1111/j.1365-2958.2007.05643.x

5. Fus3 can affect phosphorylation of TFs directly or indirectly. However, the current results do not provide any important information about the direct targets of Fus3.

Response:

Thank you for your comments. Based on our phosphoproteome data, we noticed that several TFs showed the down-regulated phosphorylation levels in $\Delta fus3$, indicating these TFs might be directly or indirectly regulated by Fus3 at phosphorylation levels. At first, we summarized the down-regulated phosphorylation TFs to investigate the Fus3 regulations and functions.

Several TFs have been regarded as the direct target of Fus3, such as VeA, Con7, and SteA (Odenbach et al., 2007; Bayram et al., 2012; Katayama T et al., 2021). Deletion of *fus3* leads to the down-regulated phosphorylation level, and furtherly results in the decreased transcription of downstream genes and the defect development phenotype. And according to the potential targets, we predicted that the phosphorylation motif of Fus3 might be the [RxxSP] in *A. flavus*.

Importantly, we also identified that AccA, not as a TF, might be potential target of Fus3. The phosphorylation level of AccA was down-regulated in $\Delta fus3$, the direct interaction of AccA and

Fus3 were noticed by Fus3-TAP analysis, and the phosphorylation motif [RxxSP] was also identified in AccA. All information suggested that AccA could be a direct target of Fus3. Furtherly, the phosphorylation level of AccA affects ACCase activity (Jia et al., 2017). In our research, the ACCase activities in Fus3 defective strains were significantly decreased, while the ACCase activity of *fus3*^{T182D} was partly recovered (Fig. 6G). So, in our research, Fus3 could directly regulate the phosphorylation of AccA, and then affect the ACCase activity and malonyl-CoA level, subsequently positively modulate the AFs production. This is the mainly conclusion in our manuscript.

In addition, we also noticed the down-regulated phosphorylation level of other TFs, such as AtfA, AtfB, AP-1. Previous reports showed that phosphorylation levels are critical for their transcriptional activated functions (Choi, 2014; Sanchez-Mir et al., 2020). But in our research, Fus3 regulated AFs independent on AFs cluster genes and these regulators. So, we did not perform the further investigations, and we moved these TFs information to the supplementary materials in the revised manuscript (Table S2). We might discover more Fus3 targets and study more Fus3 regulations in future research.

Wei J, Zhang Y, Yu TY, Sadre-Bazzaz K, Rudolph MJ, Amodeo GA, Symington LS, Walz T, Tong L. 2017. A unified molecular mechanism for the regulation of acetyl-coa carboxylase by phosphorylation. *Cell Discovery*. **3**, 16055. doi:10.1038/celldisc.2016.44

Sánchez-Mir L, Fraile R, Ayté J, Hidalgo E. 2020. Phosphorylation of the Transcription factor Atf1 at multiple sites by the MAP kinase sty1 controls homologous recombination and transcription. *J Mol Biol*. **432**(19):5430-5446. doi:10.1016/j.jmb.2020.08.004

Choi WJ. 2014. The heterochromatin-1 phosphorylation contributes to TPA-induced AP-1 expression. *Biomol Ther (Seoul)*. **22**(4):308-313. doi:10.4062/biomolther.2014.057

January 7, 2022

Prof. Fuguo Xing
Institute of Food Science and Technology, Chinese Academy of Agricultural Sciences
Beijing
China

Re: Spectrum01269-21R2 (Fus3, as a critical kinase in MAPK cascade, regulates aflatoxin biosynthesis by controlling the substrate supply in *Aspergillus flavus*, rather than the modulation of aflatoxin biosynthetic genes)

Dear Prof. Fuguo Xing:

Your manuscript has been accepted, and I am forwarding it to the ASM Journals Department for publication. You will be notified when your proofs are ready to be viewed.

Sincerely,

Christina Cuomo
Editor, Microbiology Spectrum

Journals Department
Supplemental Material: Accept
Supplemental Material: Accept
Supplemental Material: Accept